# Three-dimensional multi-site random access photostimulation (3D-MAP)

Yi Xue[1], Laura Waller[1], Hillel Adesnik[2,3]*, Nicolas Pégard[4,5,6]*

[1]Department of Electrical Engineering & Computer Sciences, University of California, Berkeley, Berkeley, United States; [2]Department of Molecular & Cell Biology, University of California, Berkeley, Berkeley, United States; [3]Helen Wills Neuroscience Institute, University of California, Berkeley, Berkeley, United States; [4]Department of Applied Physical Sciences, University of North Carolina at Chapel Hill, Chapel Hill, United States; [5]Department of Biomedical Engineering, University of North Carolina at Chapel Hill, Chapel Hill, United States; [6]UNC Neuroscience Center, University of North Carolina at Chapel Hill, Chapel Hill, United States

**Abstract** Optical control of neural ensemble activity is crucial for understanding brain function and disease, yet no technology can achieve optogenetic control of very large numbers of neurons at an extremely fast rate over a large volume. State-of-the-art multiphoton holographic optogenetics requires high-power illumination that only addresses relatively small populations of neurons in parallel. Conversely, one-photon holographic techniques can stimulate more neurons with two to three orders lower power, but with limited resolution or addressable volume. Perhaps most problematically, two-photon holographic optogenetic systems are extremely expensive and sophisticated which has precluded their broader adoption in the neuroscience community. To address this technical gap, we introduce a new one-photon light sculpting technique, three-dimensional multi-site random access photostimulation (3D-MAP), that overcomes these limitations by modulating light dynamically, both in the spatial and in the angular domain at multi-kHz rates. We use 3D-MAP to interrogate neural circuits in 3D and demonstrate simultaneous photostimulation and imaging of dozens of user-selected neurons in the intact mouse brain in vivo with high spatio-temporal resolution. 3D-MAP can be broadly adopted for high-throughput all-optical interrogation of brain circuits owing to its powerful combination of scale, speed, simplicity, and cost.

*For correspondence:
hadesnik@berkeley.edu (HA);
pegard@unc.edu (NP)

**Competing interest:** The authors declare that no competing interests exist.

## Editor's evaluation

This paper is of interest to neuroscientists working on all-optical interrogation of neural circuits and optogenetics. It provides a new, inexpensive, one-photon approach for high-speed 3D photostimulation with sparse targeting. This new method has been well characterized and demonstrated in both in vitro and in vivo experiments on mouse brain tissue.

## Introduction

Optogenetics enables rapid and reversible control of neural activity (*Deisseroth, 2011*; *Zhang et al., 2007*). By focusing light on neurons (*Ronzitti et al., 2017*) with either one-photon (*Lutz et al., 2008*; *Anselmi et al., 2011*; *Szabo et al., 2014*; *Reutsky-Gefen et al., 2013*; *Dhawale et al., 2010*; *Leifer et al., 2011*; *Adam et al., 2019*; *Werley et al., 2017*; *Sakai et al., 2013*) or two-photon photostimulation (*Carrillo-Reid et al., 2019*; *Naka et al., 2019*; *Daie et al., 2021*; *Sridharan, 2021*; *Nikolenko et al., 2008*; *Papagiakoumou et al., 2010*; *Pégard et al., 2017*; *Gill et al., 2020*; *Marshel et al., 2019*; *Robinson et al., 2020*; *Forli et al., 2021*; *Mardinly et al., 2018*), one can

elicit or suppress the activity of custom neural ensembles in order to map neural circuits (*Dhawale et al., 2010*; *Leifer et al., 2011*; *Naka et al., 2019*; *Daie et al., 2021*; *Sridharan, 2021*) and draw links between specific patterns of neural activity and behavior (*Szabo et al., 2014*; *Reutsky-Gefen et al., 2013*; *Adam et al., 2019*; *Carrillo-Reid et al., 2019*; *Gill et al., 2020*; *Marshel et al., 2019*; *Robinson et al., 2020*). Two-photon photostimulation has the advantages of high spatial resolution for precise neural activity control and relative immunity to tissue light scattering, enabling stimulation of deep brain circuits. However, two-photon holographic optogenetic photostimulation can only stimulate relatively small ensembles of neurons at a time, limited by the very high-power pulsed illumination required to achieve non-linear multiphoton absorption. High photon density heats brain tissue, and can disturb brain activity and cause thermal damage (*Mardinly et al., 2018*; *Podgorski and Ranganathan, 2016*). The accessible volume for holographic two-photon photostimulation is also limited by the coherence length of laser being used, since holographic diffraction of femtosecond pulses to a large angle introduces severe pulse dispersion ('chirp'), substantially reducing pulse peak power (*Sun et al., 2019*). Mechanically scanning a single focus or a holographic pattern can access a larger volume but is not suitable for studies that require simultaneous illumination (*Yang et al., 2015*) to understand the network. Perhaps the two most significant barriers to the broader adoption of multiphoton optogenetic systems is that they are extremely expensive and sophisticated to operate and to maintain. Thus, despite more than a decade since their introduction, multiphoton optogenetics has only been adopted by a small handful of groups, typically with much prior optical expertise.

In contrast, one-photon photostimulation systems require two to three orders of magnitude less laser power to activate neurons. They are much simpler to implement with far less expensive hardware, and they can still achieve high-resolution photostimulation under limited light scattering conditions. There are three main strategies to generate multiple foci with one-photon photostimulation, yet none is capable of simultaneously stimulating custom ensembles of multiple neurons over a large 3D volume with high spatial resolution. The first approach, scanning-based one-photon photostimulation, stimulates neurons sequentially by rapidly scanning a single focus across small neural clusters with scanning mirrors (*Petreanu et al., 2009*) or acousto-optic deflectors (AODs) (*Losavio et al., 2010*; *Wang et al., 2011*). This method cannot photostimulate distributed ensembles simultaneously. The second approach is to directly project 2D illumination patterns onto samples with a digital micromirror device (DMD) (*Dhawale et al., 2010*; *Leifer et al., 2011*; *Adam et al., 2019*; *Werley et al., 2017*; *Sakai et al., 2013*). This widefield illumination scheme has a large field-of-view (FOV) and moderate lateral resolution but only modulates light in 2D. There is no axial sectioning ability and unwanted photostimulation may occur both above and below the focal plane. The third approach, computer generated holography (CGH), generates 3D distributed foci by phase modulation of coherent light in Fourier space using a spatial light modulator (SLM) (*Lutz et al., 2008*; *Anselmi et al., 2011*; *Szabo et al., 2014*; *Reutsky-Gefen et al., 2013*). One-photon CGH has high lateral resolution and moderate axial resolution but has several critical drawbacks that constrain its ability to execute sophisticated optogenetics experiments. First, the throughput of CGH that determines the number of accessible voxels is fundamentally limited by the number of degrees-of-freedom (DoF), which is determined by the number of pixels and the bit depth of the SLM, regardless of magnification and numerical aperture. This generally results in a small FOV when high spatial resolution is required. Second, the refresh rate (tens to hundreds Hz) of SLMs limits the speed of CGH. Third, CGH requires computing a 2D phase mask at the Fourier plane for a 3D hologram pattern, which is an ill-posed problem that requires iterative optimization, with computation time on the order of minutes (*Zhang et al., 2017*; *Gerchberg, 1972*; *Leseberg, 1992*; *Xue et al., 2019*) when shaping light across multiple z-planes. Although recent work successfully reduces the CGH computation time to milliseconds using a pre-trained deep neural network (*Hossein Eybposh et al., 2020*), this will be insufficient when the number of z-planes increases. The extra computational requirements for generating holographic patterns in the Fourier domain (rather than directly projecting patterns on the conjugate image plane) can become limiting when thousands of different patterns are required (as in high-throughput mapping experiments), or when fast online synthesis of custom patterns is needed for closed-loop experiments. The fourth drawback of one-photon CGH is that holograms composed of many illumination spots will suffer spatial cross-talk as out-of-focus light from each focused spot interacts, accidentally stimulating non-targeted neurons. Taken together, despite the power of these previous one-photon techniques, none

are suitable for large-scale high-resolution optogenetic activation of 3D distributed ensembles of neurons.

To overcome this challenge, we developed 3D multi-site random access photostimulation (3D-MAP), a new approach to generate 3D illumination patterns by modulating light both in the angular domain, $(k_x, k_y)$, with scanning mirrors, and in the spatial domain, $(x, y)$, with a DMD. For 3D optogenetic photostimulation, illumination patterns are optimized to target an opsin expressed in the neuronal soma (~10 μm). The set of light rays needed to generate each spherical target can be described with a 4D light field, $(x, y, k_x, k_y)$, in the spatio-angular domain. 3D-MAP generates these rays by rapidly sweeping through the appropriate angles of illumination with scanning mirrors while using the DMD to project the corresponding amplitude masks to pattern each angle's spatial information. 3D-MAP uses one spatial pattern on the DMD for each unique illumination angle set by the scanning mirrors. The total number of DoF is hence determined by the *product* of the number of angles of the scanning mirror and the number of pixels of the DMD (as opposed to the *sum*), resulting in a much larger DoF than existing one-photon optogenetic stimulation techniques (*Lutz et al., 2008*; *Anselmi et al., 2011*; *Szabo et al., 2014*; *Reutsky-Gefen et al., 2013*; *Dhawale et al., 2010*; *Leifer et al., 2011*; *Adam et al., 2019*; *Werley et al., 2017*; *Sakai et al., 2013*). Thus, 3D-MAP achieves both high spatial resolution and a large accessible volume. Compared to one-photon CGH, 3D-MAP is able to reduce spatial cross-talk by prioritizing illumination angles that minimize the stimulation of non-targeted areas due to its high DoF. Compared to 2D widefield patterning methods, 3D-MAP retains the advantages of high-throughput and computational efficiency, while extending the addressable space from 2D to 3D. Thanks to the use of a DMD instead of an SLM, we can pattern the entire addressable volume at multi-kHz rate (the volumetric pattern refresh rate), which exceeds the bandwidth of most neural circuits and is one order of magnitude faster than 3D CGH. A comparison between strengths and weaknesses of 3D-MAP versus existing photostimulation approaches is shown in (*Appendix 1—table 1*).

We present the experimental setup and computational methods for all-optical interrogation of neural circuits in 3D and demonstrate that 3D-MAP achieves high-resolution, high-speed, and high-throughput in brain slices and in vivo recorded by electrophysiology and optical detectors, respectively. We then use 3D-MAP to interrogate neural circuits with both single-spot photostimulation and multi-spot photostimulation in 3D. These experiments validate 3D-MAP as a one-photon technique to manipulate neural circuits on-demand with high spatio-temporal resolution in the intact brain. Our technique can be flexibly used to map neural connectivity at both small and large scales. Its relative simplicity, small hardware footprint, and lower cost should make it broadly adoptable across the neuroscience community.

## Results

### Optical design of 3D-MAP

The experimental setup for 3D-MAP is shown in *Figure 1*. A DMD modulates amplitude in real space, $(x, y)$, while scanning mirrors control the angles of illumination, $(k_x, k_y)$. Both devices are placed at conjugate image planes and time-synchronized (see Materials and methods for details). We first compute the desired intensity of the light field $(x, y, k_x, k_y)$, and then sequentially display the amplitude mask pattern on the DMD as the scanning mirrors sweep through the relevant angles (*Figure 1B*). For example, in the simplest case of patterning a single focus spot at the native focal plane (z = 0), we simply turn on the pixels corresponding to the desired position, keeping the pattern constant for all illumination angles. To create a focused spot at an off-focus plane, a small aperture on the DMD rotates in a circle as the scanning mirrors sweep through a cone of projection angles (*Figure 1B*). The center of the aperture's revolving circle is the lateral position of the target spot, and the circle's diameter, *D*, determines the spot's axial position (a larger diameter concentrates light further away from the native image plane at a given illumination angle). To activate multiple neurons simultaneously, we superimpose the patterns corresponding to each target (*Figure 1D*). In addition, we can adjust the laser power in each focused spot flexibly and individually by adjusting the aperture size on the DMD, the number of apertures aimed at a given target, or by adjusting the laser power if needed. The amount of light on any given target is quadratically proportional to the radius of the aperture. For

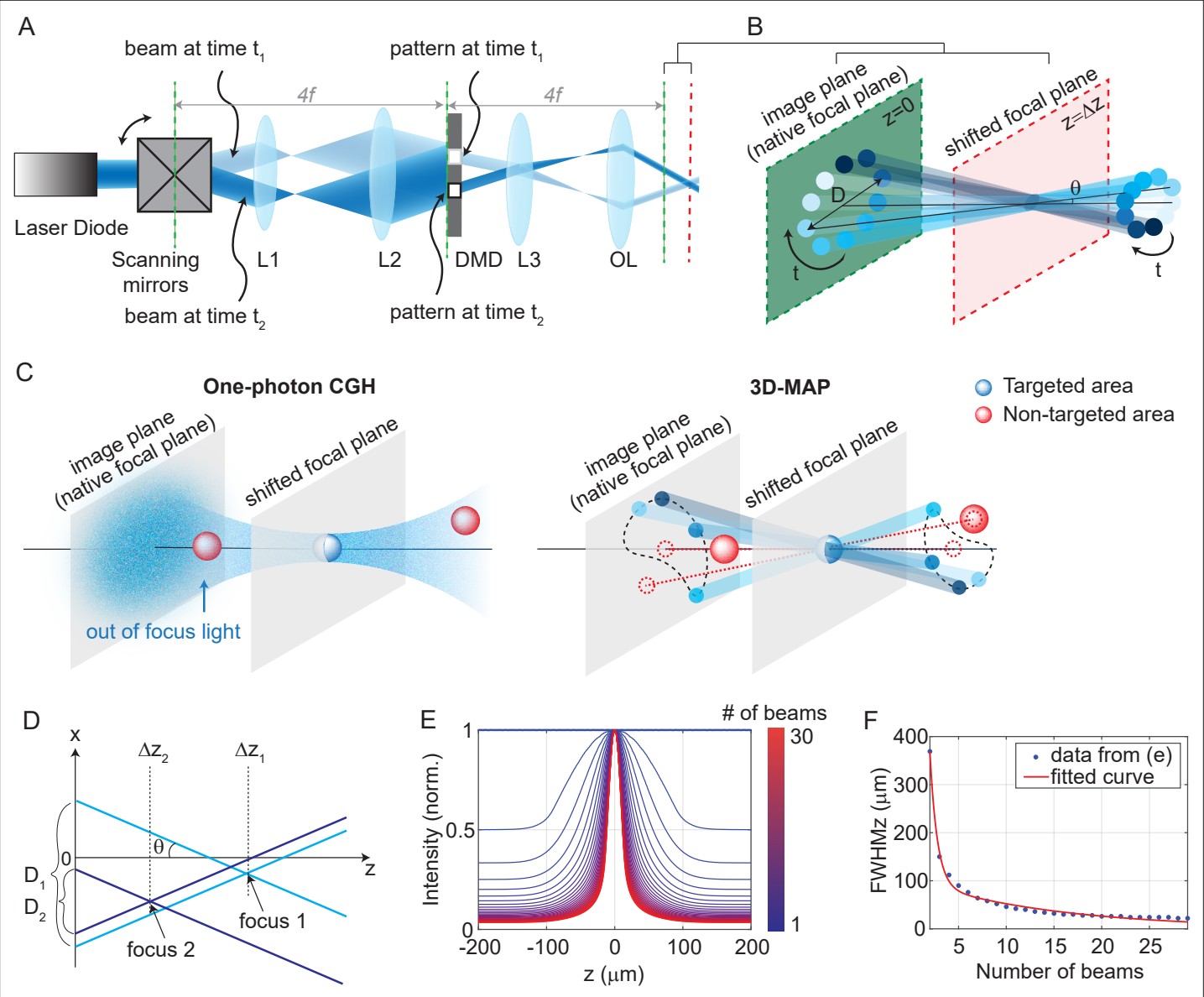

**Figure 1.** Experimental setup for three-dimensional multi-site random access photostimulation (3D-MAP). (**A**) A collimated laser beam illuminates the surface of a digital micromirror device (DMD) with a custom illumination angle set by scanning mirrors. The DMD is synchronized with the scanning mirrors to match the 2D mask of the spatial aperture to the illumination angle. (**B**) Detailed view of the overlapping amplitude masks and illumination angles at the conjugate image plane (green) showing how synchronized illumination angles and amplitude masks can generate a focused spot away from the native focal plane (red). Circular patterns labeled by different colors are spatial apertures projected at different times. The position illuminated by all beams while sweeping through each illumination angle forms a focus at the shifted focal plane at $z=\Delta z$. $D$ is the diameter of the sweeping trace. $\theta$ is the illumination angle. (**C**) A focus generated by computer generated holography (CGH) stimulates the targeted area (blue) in focus but also stimulates non-targeted areas (red) out of focus. 3D-MAP can stimulate only the targeted areas and avoid non-targeted areas by closing the amplitude apertures along propagation directions that project to non-targeted areas (dashed red line). (**D**) Multiple foci can be generated simultaneously at various depths by superposition of their perspective projection along each illumination angle. (**E**) Simulated maximum intensity profile along the z-axis for an increasing number of overlapping beams shows how axial resolution increases with the number of superimposed projection directions. (**F**) Full-width-half-maximums (FWHMs) of the illumination patterns in (**E**). The data for a single beam is excluded because it has no axial sectioning ability.

The online version of this article includes the following figure supplement(s) for figure 1:

**Figure supplement 1.** Comparison of three-dimensional (3D) spatial resolution of 3D-multi-site random access photostimulation (MAP) versus existing photostimulation approaches.

**Figure supplement 2.** Three-dimensional multi-site random access photostimulation (3D-MAP) can automatically remove illumination angles in non-targeted areas.

example, changing the aperture radius from 10 pixels to 14 pixels will increase the stimulation power on the target by a factor 2 times.

Another way to describe the pattern design is that the amplitude masks are computed by tracing the perspective views of the target back to the objective's native focal plane along parallel ray directions set by the scanning mirrors. Hence, even complicated 3D target designs can be computed quickly and easily. In contrast, conventional CGH systems synthesize a 3D hologram by modulating a coherent laser beam with a 2D static phase mask at the conjugate Fourier plane, which is an ill-posed problem where the 3D pattern is an approximate solution. CGH uses coherent light that introduces speckle in the hologram pattern (*Figure 1C*, *Figure 1—figure supplement 1A-D*), and out-of-focus light above and below each targeted area generates cross-talk that may unintentionally activate non-targeted neurons. Conversely, 3D-MAP modulates light incoherently in time which eliminates speckle. The cross-talk can be reduced by not selecting certain illumination angles to avoid stimulating non-targeted areas, further enhancing the stimulation accuracy and efficiency (*Figure 1C*). Our algorithm can automatically remove rays in the non-targeted areas and generate the corresponding DMD amplitude patterns (an example is shown in *Figure 1—figure supplement 2*). More strategies to reduce cross-talk are discussed in the Discussion section.

The axial resolution of 3D-MAP depends on the angle and number of beams that are sequentially projected to generate the targeted 3D intensity pattern (*Figure 1E–F*). Because the fastest frame refresh rate of the DMD (13 kHz, or 77 µs per frame) is about 52-fold shorter than the stimulation time (4 ms, see later sections for details), we can use up to 52 angles without sacrificing speed. However, the DMD has limited on-chip memory, so we choose to minimize the number of amplitude masks required. In all our experiments, we use 10 illumination angles for each 3D target; additional illumination angles only introduce minor improvements (*Figure 1E–F*). The maximum refresh rate of the DMD is 13 kHz, and each 3D volumetric pattern is generated by 10 masks on the DMD, so the maximum volumetric pattern rate is 1.3 kHz, which is more than an order of magnitude faster than conventional CGH with commercial SLMs. Since the characteristic response time of the microbial opsin is much longer than the duration of individual projection masks, the relevant light sculpting pattern seen by the opsin is the time-averaged sum of the intensity of several mutually incoherent masks. In addition, the DMD masks for 3D-MAP are based on ray tracing and can be calculated much faster than phase holograms in CGH (*Zhang et al., 2017*; *Gerchberg, 1972*; *Leseberg, 1992*; *Xue et al., 2019*), enabling real-time or closed-loop applications such as mapping neural circuits (see below).

## Optical characterization of 3D-MAP

To quantify the optical resolution of 3D-MAP, we first measured the 3D optical point spread function (PSF). We turned on a 1-pixel aperture on the DMD (acting as a point source) and scanned through angles by scanning mirrors to generate a focus, which is imaged by a sub-stage camera with a thin fluorescent film on a microscope slide at many depth planes (*Figure 2A–C*). The full-width-half-maximums (FWHMs) of the resulting 3D optical PSF indicate a spatial resolution of $5 \times 5 \times 18$ µm$^3$ with a 473 nm excitation wavelength. This PSF is measured in the center of the FOV and represents the best optical resolution of 3D-MAP, while the resolution will degrade when the focus is not paraxial. We also demonstrated the ability to simultaneously generate 25 foci at custom (x, y, z) locations in a $744 \times 744 \times 400$ µm$^3$ volume (*Figure 2D*). Additional examples of simultaneous 3D multi-spot generation are shown in *Figure 2—figure supplement 1A-B*. The foci on the edge of the accessible volume are slightly larger than the foci in the center generated by the same size aperture (63% larger along the x-axis and 38% larger along the z-axis on average for foci that are 300 µm away from the center, *Figure 2—figure supplement 1C-J*). Together, these data show that 3D-MAP achieves high spatial resolution for multi-site stimulation over a large volume.

We next quantified the effects of optical scattering in mouse brain tissue on focusing capabilities with 3D-MAP by measuring the size of a target after propagating through acute brain slices (brain slices that are kept vital in vitro for hours) of increasing thickness placed just above the thin fluorescent film (*Figure 2E*). We generated a focused target by turning on a 10-pixel radius aperture on the DMD (8 µm in diameter at the native focal plane) to produce an illumination volume matching the typical dimensions of a neuronal soma in the cortex. We note that this target size is larger than the spatial resolution limit of 3D-MAP, but that it represents a practical choice of 3D pattern that matches the size of neurons and our application. The aperture size for all experiments is listed in (*Appendix 1—table 3*)

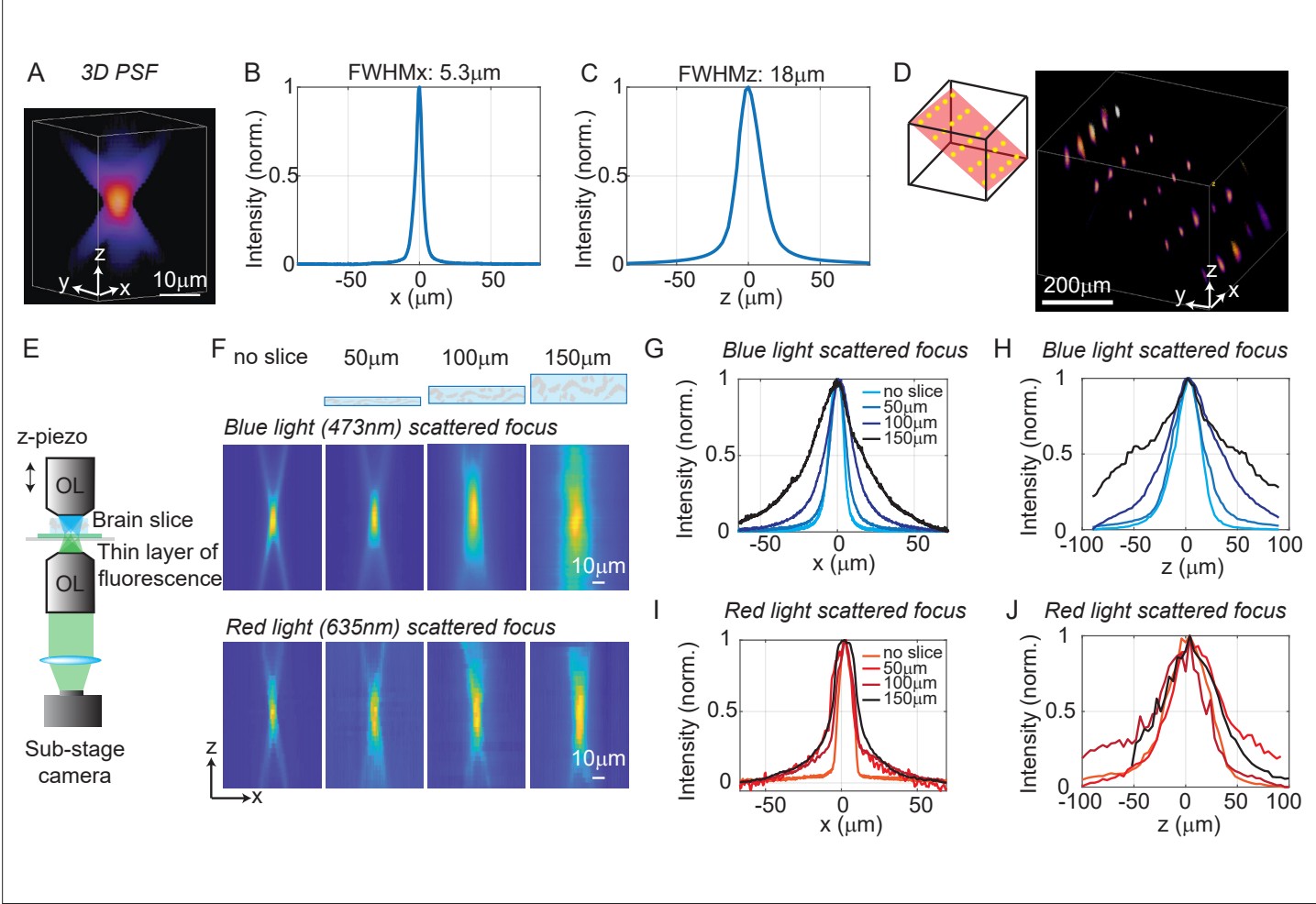

**Figure 2.** Optical characterization of the spatial resolution of three-dimensional multi-site random access photostimulation (3D-MAP) under increasing optical scattering conditions. (**A**) Experimentally measured 3D optical point spread function (PSF) (1-pixel aperture on DMD) built from focus-stacked 2D images of a thin, uniform fluorescent calibration slide recorded at different depths using a sub-stage camera. (**B**) The PSF's lateral cross-section (x-axis) has a full-width-half-maximum (FWHM) of 5.3 μm. (**C**) The PSF axial cross-section (z-axis) has an FWHM of 18 μm. (**D**) Left: we simultaneously generated 25 foci within a 744 × 744 × 400 μm³ volume. Right, experimental measurement of the corresponding 3D fluorescence distribution. (**E**) Schematic diagram of the sub-stage microscope assembly for 3D pattern measurement. (**F**) XZ cross-section of the PSF, measured with blue (473 nm) and red (635 nm) light stimulation without scattering, and through brain tissue slices of increasing thickness: 50, 100, and 150 μm. (**G**) Under blue light illumination, the FWHM along the x-axis for increasing amounts of scattering is 11.7, 12.2, 19.7, and 29.0 μm, and (**H**) the FWHM along the z-axis is respectively 42, 46, 76, and 122 μm. (**I**) With red light illumination, the FWHM along the x-axis for increased amounts of scattering is 10.4, 19.3, 26.7, and 29.6 μm, and (**J**) the FWHM along the z-axis is respectively 50.7, 75.4, 79.7, and 73.1 μm.

The online version of this article includes the following figure supplement(s) for figure 2:

**Figure supplement 1.** Three-dimensional multi-site random access photostimulation (3D-MAP) is able to simultaneously generate multiple foci anywhere in 3D.

('Summary of experiment details'). We compared scattering for both blue (473 nm) and red (635 nm) excitation wavelengths. Without the brain slice, the size of the focus was 11.7 × 11.7 × 42 μm³ (473 nm blue) and 10.4 × 10.4 × 50.7 μm³ (635 nm red). When generating the focus through 50, 100, and 150 μm thick brain slices, the size of the focus spot with blue light increased by 8%, 75%, and 269%, and the size of the focus spot with red light increased by 67%, 107%, and 114%, respectively (averaged values measured along the x-axis and the z-axis). Compared to other one-photon photostimulation techniques (*Szabo et al., 2014*; *Sakai et al., 2013*), 3D-MAP achieves high optical resolution especially when the scattering is weak (in brain slices or in superficial layers in vivo), and the resolution can be improved in the future by using red stimulation light combined with red-shifted opsins such as Chrimson or ChRmine (see Discussion).

## 3D-MAP photostimulation in brain tissue

We next quantified the physiological spatial resolution as measured by the physiological point spread function (PPSF) of 3D-MAP by stimulating neurons in acute mouse brain slices and in vivo. We performed whole-cell patch clamp recordings from L2/3 excitatory neurons expressing a soma-targeted version of the potent optogenetic protein ChroME (*Mardinly et al., 2018*) (see Materials and methods). The PPSF measures the photocurrent response as a function of the radial and axial displacements between the targeted focus and the patched cell (*Pégard et al., 2017*). The physiological spatial resolution is not only related to the optical PSF but also to the laser power needed to drive a neuron to spike (*Li et al., 2016*). Since patterns for 3D-MAP can be calculated significantly faster than with CGH (*Zhang et al., 2017*; *Gerchberg, 1972*; *Leseberg, 1992*; *Xue et al., 2019*), we measured the volumetric PPSF by recording the photocurrent response from 2541 targeted locations (a 11 × 11 × 21 pixels grid) in 2–5 min, limited only by opsin kinetics. We chose a 4 ms stimulation time (also known as 'dwell time') in all experiments to ensure full activation of opsins (*Figure 3—figure supplement 1*), followed by 8–40 ms inter-stimulation time (time between two stimulation pulses in

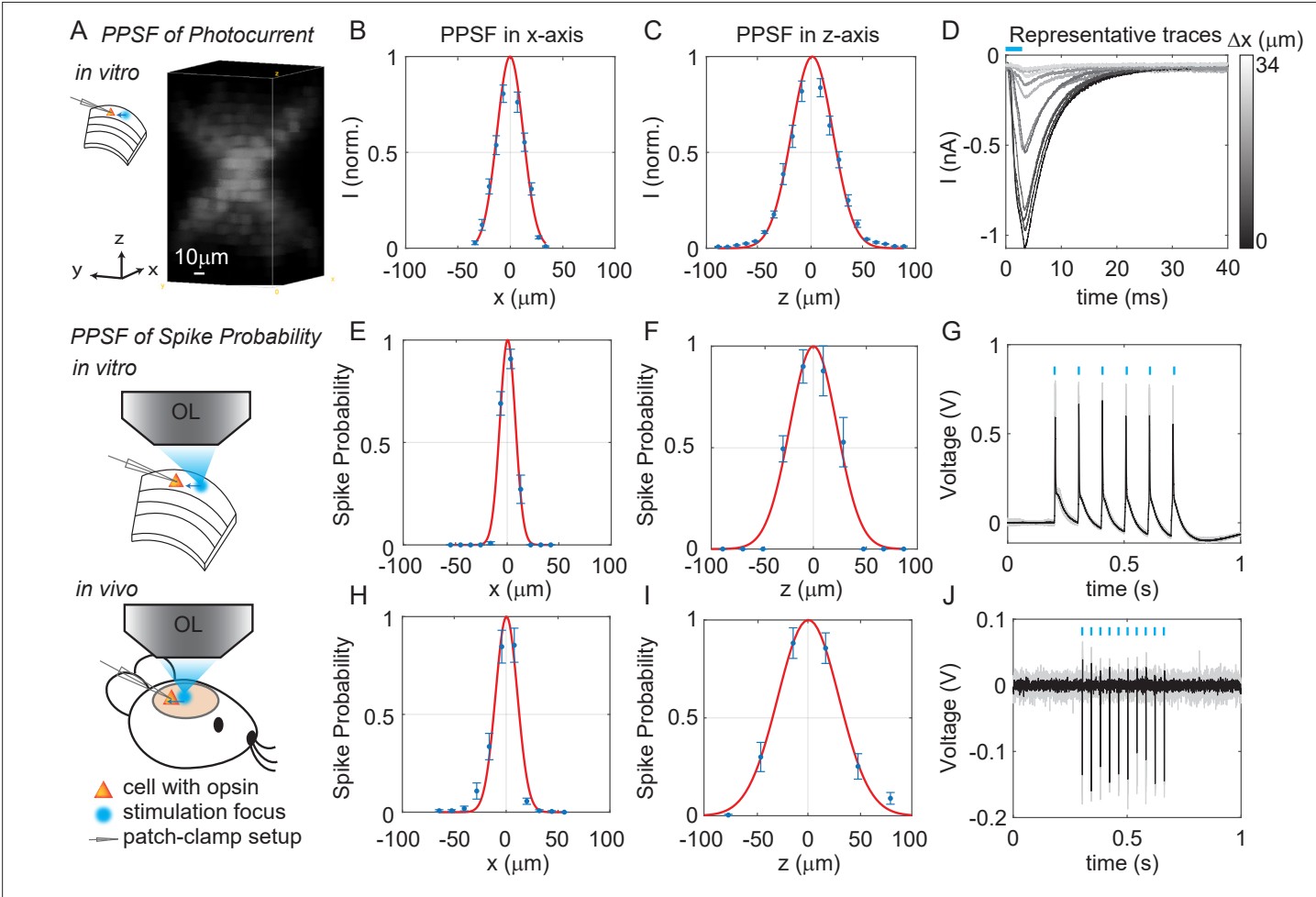

**Figure 3.** Three-dimensional multi-site random access photostimulation (3D-MAP) enables high spatial resolution photo-activation of neurons in vitro and in vivo. (**A**) Example 3D representation of a physiological point spread function (PPSF) photocurrent measurement in vitro. (**B, C**) Photocurrent resolution (full-width-half-maximum [FWHM]) is 29 ± 0.8 μm laterally and 44 ± 1.6 μm axially (n = 9 neurons). (**D**) Representative traces of direct photocurrent from 11 positions along the x-axis without averaging. (**E, F**) 3D-MAP evoked spiking resolution in brain slices is 16 ± 2.4 μm laterally and 44 ± 8.9 μm axially (n = 5 neurons). (**G**) Representative traces of spike probability of 1 for in vitro measurements. (**H, I**) 3D-MAP evoked spiking resolution measured in vivo is 19 ± 3.7 μm laterally and 45 ± 6.1 μm axially (n = 8 neurons). (**J**) Representative traces of spike probability of 1 for in vivo measurements. The data shows the mean ± s.e.m. (standard error of the mean) for plots B–C, D–F, and H–I.

The online version of this article includes the following figure supplement(s) for figure 3:

**Figure supplement 1.** Temporal response of (**A–D**) ChroME and (**E–H**) ChRmine, expressed in CHO cells.

successive targets, depending on the opsin type) to avoid photocurrent accumulation and minimize short-term plasticity of the synapses under study. Note that this is a specific choice for our experiments rather than the speed limit of 3D-MAP. The PPSF measurements show that 3D-MAP provides high-resolution photostimulation in vitro (lateral FWHM, 29 ± 0.8 µm; axial FWHM, 44 ± 1.6 µm, 9 cells, *Figure 3A–D*). Under these conditions, we found that the spatial specificity of 3D-MAP is only approximately two-fold worse when compared to our previous results using multiphoton optogenetics in comparable conditions (*Pégard et al., 2017*; *Mardinly et al., 2018*). However, the PPSF of photocurrents for 3D-MAP should degrade more steeply as a function of tissue depth compared to multiphoton optogenetics due to scatter (see details in Discussion).

To quantify the physiological resolution of 3D-MAP for supra-threshold neuronal activation, we also measured the spiking probability of neurons in acute brain slices and in vivo along the lateral and axial dimensions. We targeted opsin-expressing neurons in the upper 100 µm of the brain for the in vivo PPSF measurements. While patching the neuron, we digitally displaced the target generated by 3D-MAP along the x-axis and z-axis and recorded the number of spikes. Experimental results show that 3D-MAP enables high spatial specificity under all these conditions (*Figure 3E–G* shows in vitro results [5 cells], with lateral FWHM, 16 ± 2.4 µm; and axial FWHM, 44 ± 8.9 µm. *Figure 3H–J* shows in vivo results (8 cells), with lateral FWHM, 19 ± 3.7 µm; and axial FWHM, 45 ± 6.1 µm). This PPSF is to show the best measurement in vivo, which is expected to degrade as going deeper inside of the tissue.

## Synaptic connectivity mapping over large volumes by a single focus

In addition to high spatio-temporal resolution and fast computational speed, 3D-MAP operates at low stimulation powers far below photodamage thresholds and is hence easily scalable to address large neural ensembles in distributed volumes of brain tissue. To demonstrate these advantages, we first used 3D-MAP to probe synaptic connectivity in 3D by randomly scanning a single focus point (*Figure 4*). We expressed soma-targeted ChroME in excitatory neurons of the cortex (see Materials and methods, *Appendix 1—table 3*Table S3) and performed whole-cell voltage clamp recordings from inhibitory interneurons that do not express ChroME under these conditions, to avoid the confounding effect of direct photocurrents in the patched neuron (*Figure 4A*). The widefield fluorescent image (*Figure 4B*) shows the excitatory neurons (in red) and the patched inhibitory interneuron (in green). We first mapped an 800 × 800 µm² FOV at low spatial sampling (40 µm grid) to identify the sub-regions that contained most of the presynaptic neurons (*Figure 4C*). We then re-mapped these sub-regions at fine resolution (20 µm/pixel, *Figure 4D*) and again at even higher resolution (9 µm/pixel, *Figure 4E*). Due to variable opsin protein expression levels across neurons, as well as variable intrinsic neural excitability, neurons are differentially sensitive to light. By mapping the same sub-region at different power levels, we were able to take advantage of this variability of photosensitivities to help identify putative individual (i.e., 'unitary') sources of presynaptic input (*Figure 4D*). We generated excitatory postsynaptic currents (EPSCs) by sequential photostimulation of the entire volume and observed that most connectivity maps exhibited spatial clusters (*Figure 4D–E*) as typically observed when mapping synaptic input in space (*Naka et al., 2019*). The size of each cluster is about 30 µm, which reaches the limit of spatial resolution measured in *Figure 3A–B*. Therefore, we cannot distinguish whether the cluster contains a single presynaptic neuron or not. However, within each cluster, most of the postsynaptic responses had similar amplitudes and time courses, suggesting they primarily arose from just one or a small number of presynaptic neurons (*Figure 4F–G*). We also performed synaptic connectivity mapping in vivo (*Figure 4—figure supplement 1*). These results demonstrate that 3D-MAP is easily scalable and suitable for obtaining high-resolution and large-scale connectivity maps of neural circuits in 3D.

One of the major advantages of 3D-MAP over conventional single point scanning approaches (*Petreanu et al., 2009*; *Losavio et al., 2010*; *Wang et al., 2011*) is that it has the capacity to simultaneously stimulate multiple neurons distributed anywhere in the addressable volume. Multi-site photostimulation is crucial for perturbing or mapping brain circuits because only the activation of neural populations in parallel can drive the sophisticated activity patterns needed to understand network dynamics or behavior (*Li et al., 2016*). Thus, we next demonstrated that 3D-MAP is able to simultaneously stimulate multiple user-defined targets (*Figure 5*) that we selected according to the connection map in *Figure 4e*. While patching the same inhibitory interneuron as described above (*Figure 4*),

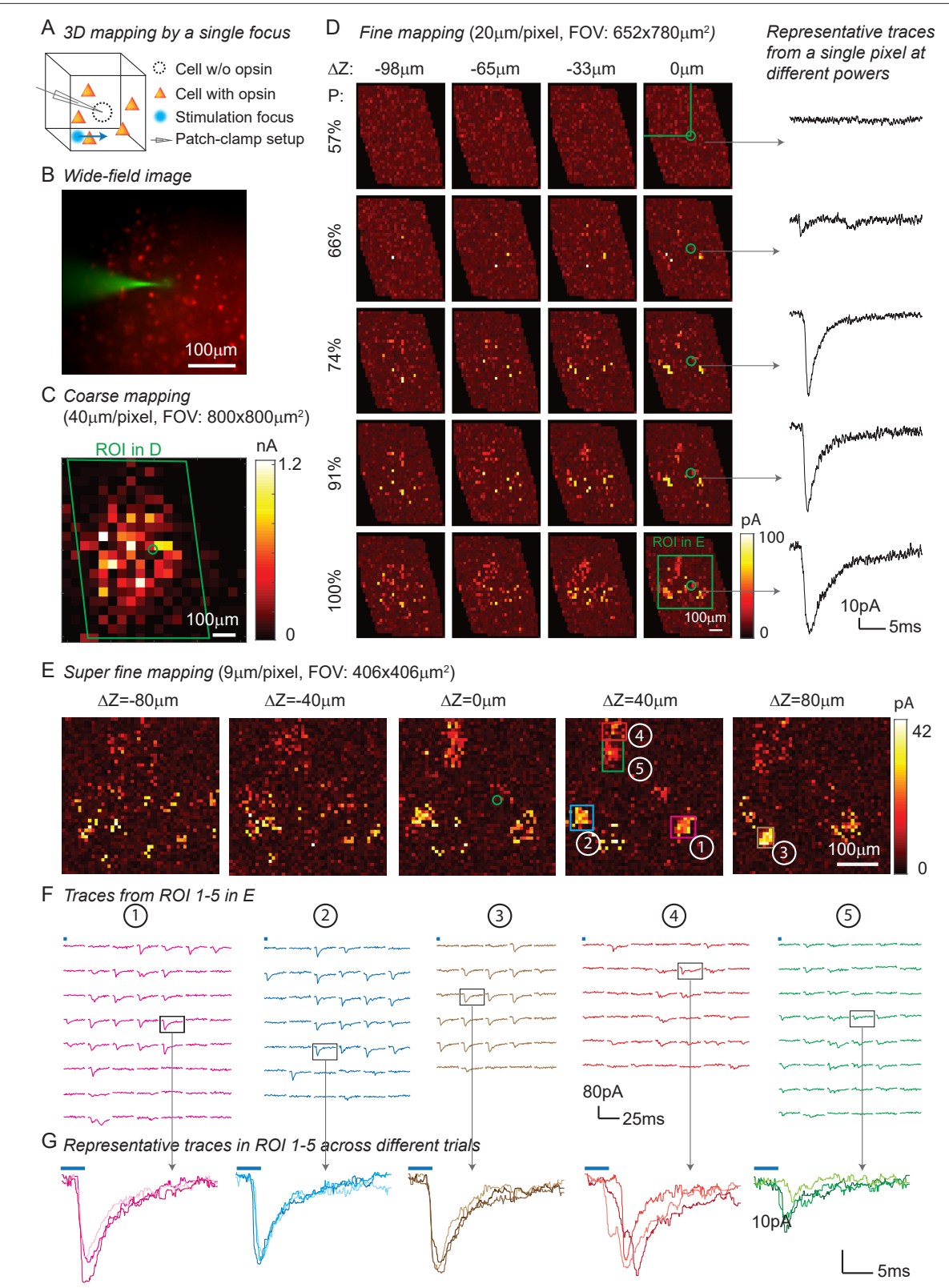

**Figure 4.** Three-dimensional (3D) mapping of excitatory synaptic connections with 3D-multi-site random access photostimulation (3D-MAP).
(**A**) Schematic diagram of the experiment. A single focus randomly scans the volume adjacent to the patched interneuron that does not express opsin, and the readout map reveals the synaptic connections between photo-activated pyramidal neurons and the patched interneuron. (**B**) An example of widefield image of opsin-expressing pyramidal neurons (red) and the patched interneuron (green). (**C**) A coarse 2D map in an 800×800 µm 2FOV at 40

*Figure 4 continued on next page*

*Figure 4 continued*

μm resolution identifies the sub-regions of the brain slice with presynaptic neurons. (**D**) Mapping the selected region at higher resolution (green box in **C**). Each row uses the same stimulation laser power 100% power: 145 μW across multiple axial planes, and each column is a map of the same axial plane at different powers. Representative excitatory postsynaptic currents EPSCs traces right show how synaptic currents at the same photostimulation pixel change as the stimulation power increases, presumably due to recruitment of additional presynaptic neurons. Data are averaged over 5 repetitions. (**E**) Super-fine resolution mapping of the region of interest (ROI) (green box in **D**) in 3D at P=90 μW. The green circle in c-e labels the location of the patched interneuron. (**F**) Traces from ROIs 1-5 labeled in E, averaged over 5 repetitions. (**G**) Representative traces of single trials without averaging from corresponding ROIs in the same color, measured across three different repetitions. The blue bar on top of traces in **F-G** indicates the 4ms stimulation time.

The online version of this article includes the following figure supplement(s) for figure 4:

**Figure supplement 1.** Three-dimensional multi-site random access photostimulation (3D-MAP) provides mapping of inhibitory synaptic connections in vivo.

we first stimulated the five presynaptic ROIs (*Figure 4E–F*) one-by-one, and then stimulated subsets of them simultaneously, and finally stimulated all the ROIs together (*Figure 5C*). We compared the photocurrents measured by multi-site simultaneous stimulation to the linear sum of individual responses by single stimulation (*Figure 5D*) and observed the multi-site stimulation generates greater net input (*Figure 5E*, p = 0.0488 for 2–5 sites stimulation, two-way analysis of variance [ANOVA]). This example shows that 3D-MAP is able to simultaneously and flexibly stimulate multiple targets and also can rapidly adjust the 3D patterns in milliseconds, which is critical for online interrogation of neural circuits. Although 3D-MAP could readily co-stimulate many more targets at a time (see the section of all-optical interrogation below), here we were limited by the number of presynaptically identified neurons.

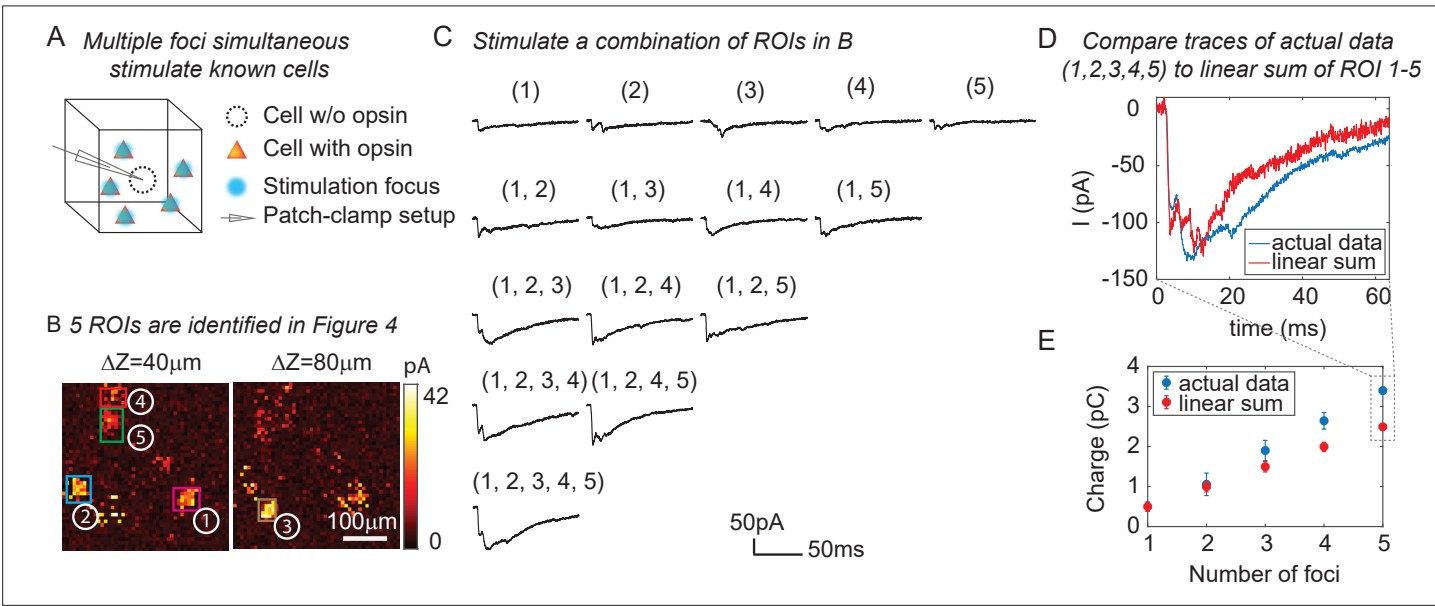

**Figure 5.** Three-dimensional multi-site random access photostimulation (3D-MAP) is able to stimulate multiple targets simultaneously to explore network dynamics. (**A**) Schematic diagram of the experiment. The stimulation ROIs are known to have synaptic connections with the patched interneuron from the widefield mapping as described in *Figure 4*. (**B**) The positions of the five ROIs are identified in *Figure 4E*. (**C**) Representative photocurrent traces for simultaneous stimulation of subsets of the five ROIs. Traces are averaged over four repetitions. The number(s) above each trace indicate the ROIs that were stimulated to generate the response. (**D**) Comparison of the actual synaptic response by simultaneous stimulation of ROIs 1–5 (blue) to the response calculated by linearly summing the traces when stimulating ROIs 1–5 individually (red). The individual response from each ROI is shown in the first row of C. (**E**) Comparison of the integral of the synaptic currents from simultaneous stimulation of multiple connected presynaptic neurons (blue) to the linear sum of the individual stimulation responses (red). The mean and standard deviation of data is calculated from all the *k*-combinations (number of foci) from the given set of five targets. The sample size is $C(5, k)$, $k = 1, 2, 3, 4, 5$.

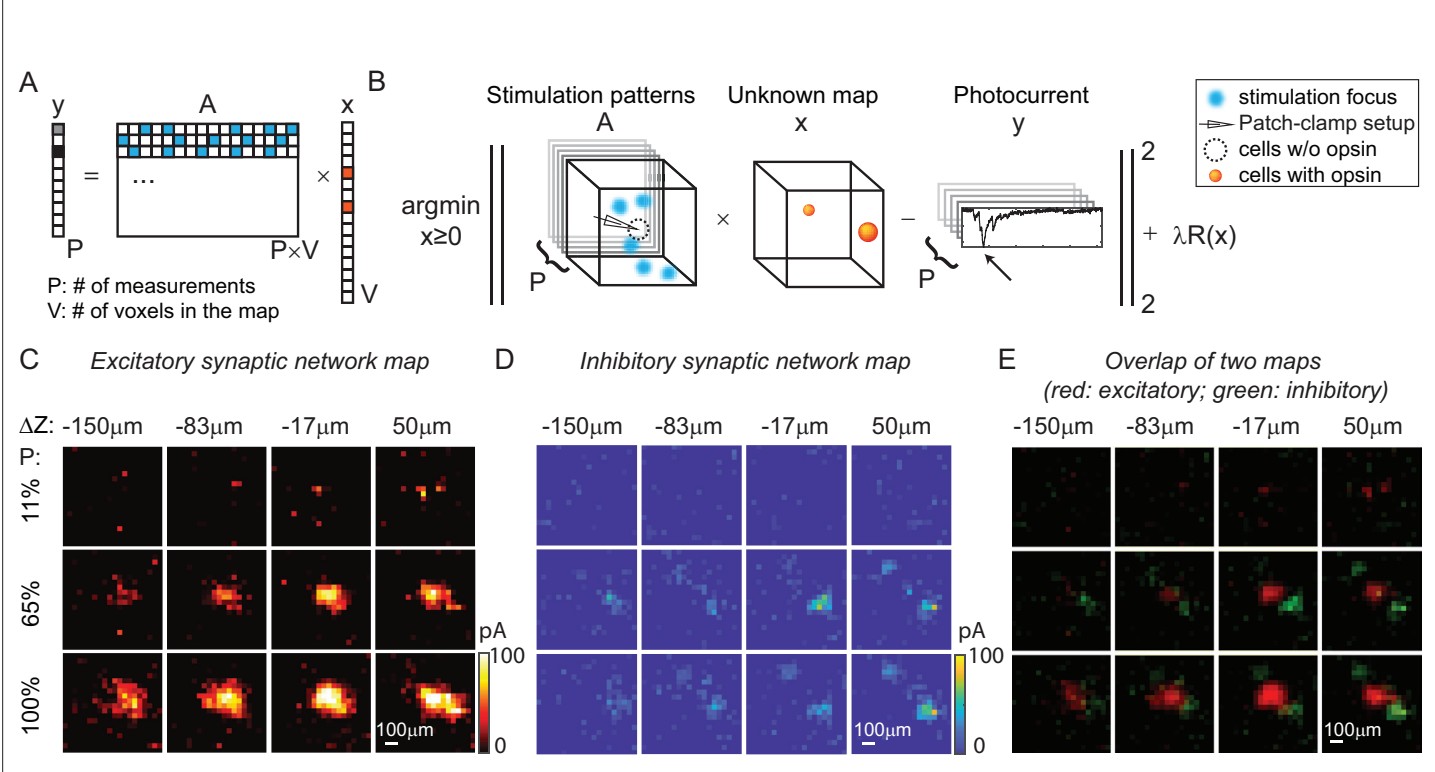

**Figure 6.** Mapping of synaptic networks in vitro by multi-site random simultaneous stimulation and computational reconstruction. (**A**) The forward model for multi-site random simultaneous stimulation. V, the number of voxels in the 3D volume. P, the number of patterns, which are orthogonal to each other. y, peak value of the measured synaptic currents. A is a matrix, where each row represents an illumination pattern, including five foci here (blue, N = 5). x is a vector of the unknown synaptic networks to be reconstructed. (**B**) Inverse problem formulation. The optimal map, x, minimizes the difference between the peak of measured currents (y) and those expected via the forward model, with a regularizer $\lambda R\left(x\right)$. (**C**) Excitatory synaptic connection map of a GABAergic interneuron located at [0, 0, 0]. (**D**) Inhibitory synaptic connection map from the same cell. (**E**) Overlap of the excitatory map (red) and inhibitory map (green) to show their spatial relationship. Figures in (**C–E**) are recorded in an 800 × 800 × 200 μm$^3$ volume at 40 μm/pixel with three different stimulation powers (100% stimulation power is 890 μW in total out of the objective lens). The number of simultaneous stimulation foci (N) is 5 in both cases and the results are average over five repetitions.

The online version of this article includes the following figure supplement(s) for figure 6:

**Figure supplement 1.** Multi-site random simultaneous stimulation by three-dimensional multi-site random access photostimulation (3D-MAP) can reconstruct synaptic connectivity maps with fewer measurements than single-target stimulation.

## Reconstructing synaptic networks by multi-site stimulation via gradient descent

Physiological brain mapping approaches must overcome the challenge of temporal throughput to map ever larger regions of brain tissue. We reasoned that mapping neural circuits with multiple foci, rather than a single focus, could scale up the temporal throughput of the system dramatically, since the overall connectivity matrix is remarkably sparse. Instead of randomly shifting a single focus to measure the one-to-one synaptic connections (*Figure 4*), we used 3D-MAP to stimulate multiple random voxels simultaneously and reconstructed the spatial map of presynaptic networks via gradient descent. In our experiment, the positions of the simultaneously stimulated voxels were randomly distributed in the 3D volume and the neural connections are unknown to begin with, unlike co-stimulating known presynaptic ROIs in *Figure 5*. In acute brain slices we expressed Chrimson in excitatory neurons of the cortex (see Materials and methods, *Appendix 1—table 3*) and again patched a GABAergic inter-neuron under voltage-clamp mode and recorded both EPSCs, and subsequently, inhibitory postsyn-aptic currents (IPSCs). We projected random sets of foci (five at a time in our experiment but other numbers should work as well, see next paragraph) and repeated this process until all voxels in 3D were stimulated several times (typically 5–10). Treating the recorded photocurrents as a combination

of responses from multiple sites, we then reconstructed the map of the synaptic network using an optimization algorithm based on gradient descent (*Figure 6A–B*, see Materials and methods).

Results in *Figure 5C–E* show the excitatory synaptic network map and inhibitory synaptic network map of the same GABAergic interneuron in an 800 × 800 × 200 µm$^3$ volume at three different stimulation powers. Multi-site simultaneous stimulation has two key advantages. First, multi-site stimulation engages the activity of spatially distributed ensembles of neurons (rather than single neurons or small local clusters), which may facilitate polysynaptic network activity and engages network level properties of the circuit. Second, since our approach is compatible with compressed sensing, it becomes possible to reconstruct the same mapping results with fewer measurements compared to single-target stimulation, assuming that the multiple stimulation sites are sparse, and that the readout signal is a linear combination of the inputs from these sites. The assumption is valid when the multiple voxels for concurrent stimulation are randomly drawn from the volume and the number of these voxels is much smaller than the total voxels in the volume (five voxels are randomly drawn from 1600 voxels in our experiment). Since multiple voxels are stimulated at the same time (say *N* voxels), each voxel needs to be measured *N* times with different patterns to estimate the contribution of each voxel to the photocurrent without compressive sensing. Under the assumption of sparsity and linearity, we can reconstruct the map of neural networks using compressive sensing and with less than *N* repetitions (*Figure 6—figure supplement 1*, *N* = 5). This feature of 3D-MAP is critical to speed up the mapping of networks in large volumes where single-target stimulation of every voxel would be prohibitively time-consuming.

## All-optical parallel interrogation of a large number of neurons in vivo

All-optical interrogation of neural circuits permits the functional dissection of neuronal dynamics in vivo and can causally relate specific patterns of neural activity to behavior. So far, this has only been possible in the living mammalian brain with two-photon holographic optogenetics (*Carrillo-Reid et al., 2019*; *Naka et al., 2019*; *Daie et al., 2021*; *Sridharan, 2021*; *Gill et al., 2020*; *Marshel et al., 2019*; *Robinson et al., 2020*). Here, we tested whether we can use 3D-MAP for all-optical interrogation of a large number of neurons in vivo, and use visible light but maintain high spatial precision. To achieve this, we added an optical detection path to 3D-MAP for fluorescence imaging (*Figure 7A*). First, we recorded a widefield image stack by mechanically scanning the objective to identify the location of labeled neurons in 3D. Then, we *simultaneously* photostimulated all the identified neurons in the volume with 3D-MAP (a pulse train of 10 pulses at 20 Hz, and each pulse is 4 ms consisting of 10 projection angles and masks), and recorded calcium activity at 10 Hz of all these neurons by widefield imaging with ROI projection using the same DMD (*Figure 7A* insert plots). Since one-photon widefield imaging has no z-sectioning ability, we took advantage of this feature to capture fluorescence signals from neurons that are above and below the focal plane (dash plane in *Figure 7A*). To improve the image contrast as well as reduce photo-bleaching, we used 2D patterns computed from the maximum z projection of the widefield image stack to selectively illuminate only the neurons (*Adam et al., 2019*) rather than the whole field when recording calcium activity. We co-expressed the blue light sensitive opsin soma-targeted stCoChR-p2A-H2B-GFP and the red calcium sensor jRCaMP1a sparsely (*Adesnik and Scanziani, 2010*) in L2/3 neurons in the mouse brain (*Forli et al., 2021*) via in utero electroporation (*Figure 7C*). The co-expression level is about 50% (sample size: 114 neurons expressing CoChR, 99 neurons expressing RcaMP1a, 54 neurons expressing both). Even though jRCaMP1a is a red fluorescent calcium indicator, it still can be excited by blue light causing a rise of fluorescence intensity. To avoid this artifact, we started imaging after stimulation of the opsin (inset plot in *Figure 7A*). The start time (*t* = 0) in all the z-scored images refers to the start time of imaging, not the start time of stimulation. We then implemented all-optical 3D-MAP to simultaneously stimulate and image, and selectively interrogate tens of neurons in the mouse brain.

We first performed a control experiment with acute brain slices to show that the observed changes in fluorescence intensity are indeed due to calcium activity (*Figure 7B*). We photostimulated and imaged 28 neurons simultaneously in vitro (see *Figure 7—figure supplement 1A* for the widefield image of the neurons) and we recorded calcium activity from these neurons before and after applying tetrodotoxin (TTX) to block action potentials. Importantly, TTX completely blocked the light-induced calcium transients demonstrating that they are due to calcium influx following light-evoked spiking. The results show obvious changes of fluorescence intensity before applying TTX and no changes after

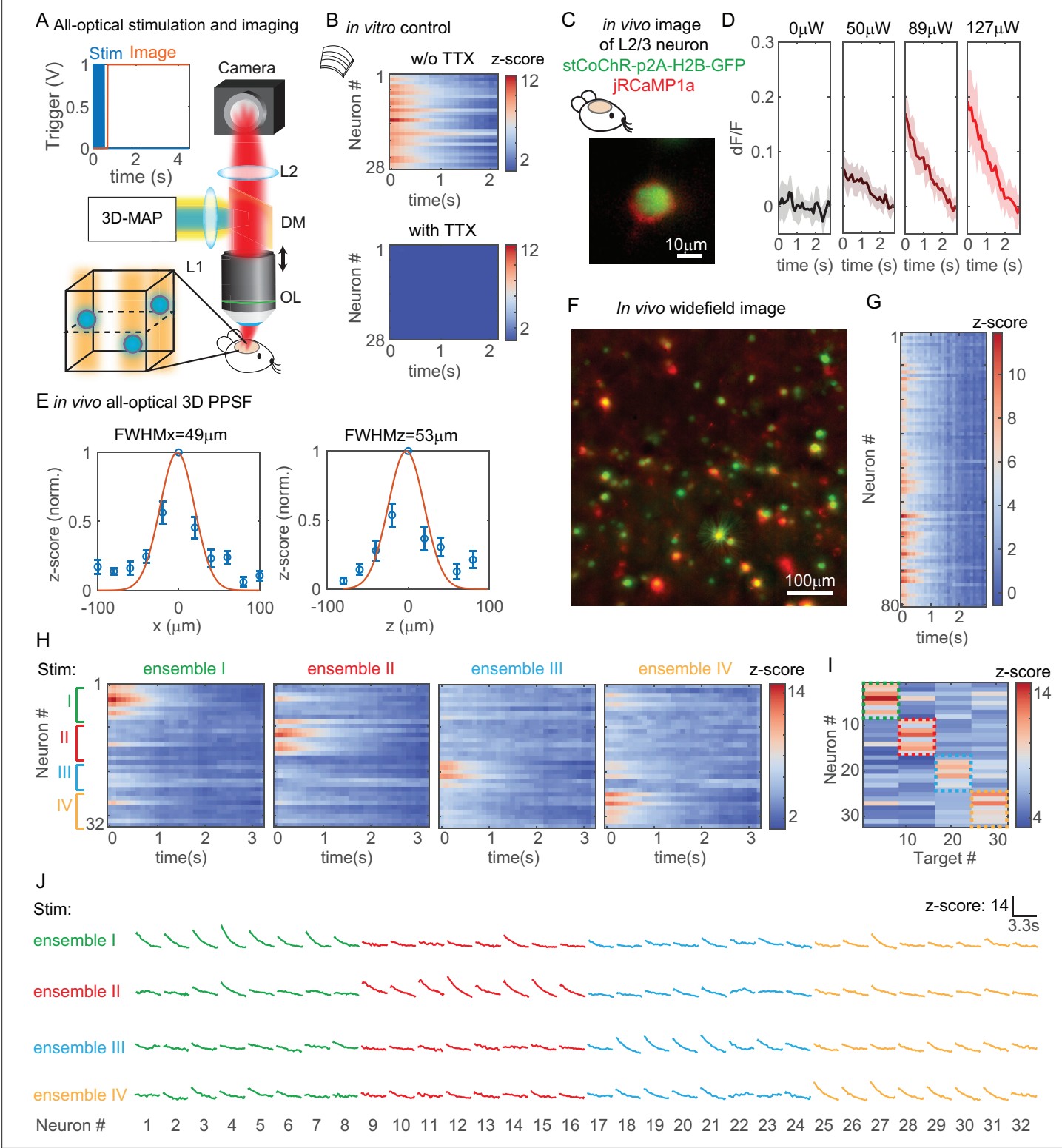

**Figure 7.** All-optical simultaneous photostimulation and imaging of groups of neurons at L2/3 in the mouse brain. (**A**) Experimental setup. The three-dimensional multi-site random access photostimulation (3D-MAP) setup as in *Figure 1A* is combined with an imaging path using a dichroic mirror, relay lenses, and a camera. Zoom-in view: neurons (blue circles) are stimulated in 3D with 3D-MAP, and calcium activity is recorded from widefield imaging with selective fluorescence excitation (yellow light) by the digital micromirror device (DMD). The dashed plane indicates the imaging focal plane. Top left: a timing plot shows that fluorescence imaging begins at *t* = 0, immediately after photostimulation. (**B**) Control experiment with brain slices. Top: calcium activity recorded from 28 neurons. Bottom: same neurons after applying TTX, when no calcium activity is detected. (**C**) Fluorescent in vivo image

*Figure 7 continued on next page*

*Figure 7 continued*

of an L2/3 neuron that co-expresses stCoChR-p2A-H2B-GFP (green) and jRCaMP1a (red). (**D**) Power test of a representative neuron, averaged across 10 repetitions. The blue box indicates the period when stimulation laser is on, and the imaging acquisition is off. (**E**) In vivo all-optical 3D physiological point spread function (PPSF). The lateral resolution is about 49 ± 21 µm and the axial resolution is about 53 ± 28 µm (*n* = 10 L2/3 neurons). (**F**) Maximum z projection of an in vivo widefield image stack (600 × 600 × 40 µm$^3$) of L2/3 neurons (green, stCoChR-p2A-H2B-GFP; red, jRCaMP1a). (**G**) Calcium activity of 80 neurons as in F recorded during simultaneous photostimulation and imaging. (**H**) Calcium activity of 32 neurons that are addressed with four distinct photostimulation patterns (labeled with different colors, also see *Figure 7—figure supplement 1H*) while fluorescence imaging data is acquired. (**I**) Peak z-score of each calcium trace recorded in H versus the corresponding stimulation patterns. The dashed colored rectangles highlight the neurons that are stimulated in each of the four patterns. (**J**) Calcium transients of 32 neurons that are stimulated with four distinct patterns, which is the line graph of the same data in (**H**).

The online version of this article includes the following figure supplement(s) for figure 7:

**Figure supplement 1.** All-optical interrogation of neurons with three-dimensional multi-site random access photostimulation (3D-MAP).

**Figure supplement 2.** The relation between optical point spread function (PSF$_o$), physiological PSF (PPSF), and all-optical PPSF (PPSF$_o$).

**Figure supplement 3.** Another example of stimulating ensembles while imaging all the neurons in vivo.

applying TTX (*Figure 7B*). Therefore, the changes of fluorescence intensity are caused by calcium activity rather than system artifacts.

All subsequent experiments were then performed in vivo in L2/3 neurons that are 200–300 µm deep (*Figure 7—figure supplement 1B*) in the intact mouse brain. We first performed a power test to determine the optimal photostimulation power needed to elicit detectable calcium activity in each neuron (*Figure 7D*). Then, we measured an in vivo all-optical 3D PPSF (a measure of calcium fluorescence change as a function of radial and axial displacements of the excitation target) with L2/3 neurons at the optimized power. The all-optical PPSF reflects the spatial specificity of all-optical 3D-MAP, incorporating the optical PSF, opsin sensitivity, and calcium indicator sensitivity, which is yet another system characterization that differs both from the optical PSF and from electrophysiology PPSF (see *Figure 7—figure supplement 2* for the relation between optical PSF, PPSF, and all-optical PPSF). Averaged measurements of all-optical PPSF across 10 neurons indicate a lateral resolution of 49 ± 21 µm and an axial resolution of 53 ± 28 µm (*Figure 7E*). An example of calcium traces from 11 positions along the x-axis in the all-optical PPSF measurement is shown in *Figure 7—figure supplement 1C*. The result demonstrates that all-optical 3D-MAP achieves high spatial resolution in vivo under the conditions used.

Next, we tested whether 3D-MAP could photo-activate a large number of neurons in vivo in 3D while maintaining spatial specificity. We photostimulated and imaged 80 user-selected L2/3 neurons in vivo simultaneously with a total power of 480 µW out of the objective lens (*Figure 7F–G*). *Figure 7F* shows the maximum z projection of a 600 × 600 × 40 µm$^3$ volume, and *Figure 7G* shows the calcium activity recorded simultaneously from every jRCaMP1-expressing neuron in the volume. The stimulation pattern is shown in *Figure 7—figure supplement 1D* and the average separation between neurons is about 49 ± 26 µm (*Figure 7—figure supplement 1E*). The number of neurons stimulated was limited here by the size of FOV and the density of expressing neurons, and potentially could be over 100 neurons with a larger FOV (see Discussion). These results demonstrate that all-optical 3D-MAP achieves high-throughput photostimulation and imaging of neurons.

Finally, we tested whether all-optical 3D-MAP can stimulate and image ensembles of spatially inter-mixed neurons as has been shown with two-photon holographic optogenetics (*Carrillo-Reid et al., 2019*; *Marshel et al., 2019*; *Mardinly et al., 2018*). The relatively sparse expression of the opsin and calcium sensor here facilitated targeting specificity (mean distance between neurons in the FOV was 67 ± 37 µm which was larger than the all-optical PPSF, *Figure 7—figure supplement 1F*). Instead of photostimulating all neurons in the volume simultaneously, we randomly assigned 32 neurons to four ensembles (eight neurons per ensemble) by Poisson disc sampling and we photostimulated each of the four ensembles sequentially while simultaneously imaging all neurons in the four ensembles with patterned illumination (*Figure 7H–J*). The widefield image of the neurons and the four stimulation patterns are shown in *Figure 7—figure supplement 1G-H*. *Figures 7H and 6J* show the calcium activity of all neurons under different photostimulation patterns, and *Figure 7I* shows the peak z-score of each neuron versus the stimulated neuron. We observe that neurons exhibit strong calcium activity primarily when they are stimulated, and minimal calcium activity when the adjacent neurons are stimulated. Another example with 30 neurons in six ensembles is shown in *Figure 7—figure supplement 3*.

These results demonstrate minimal cross-talk between photostimulated ensembles and non-targeted ensembles within the imaging volume even when stimulating and imaging L2/3 neurons in vivo. Thus, under relatively sparse opsin expression conditions, 3D-MAP can be used to photo-activate user-defined spatially intermixed ensembles in the superficial layers of the mouse brain. For densely labeled populations, 3D-MAP could still be used to photo-activate small spatial clusters of neurons which, depending on the goals of the experiments, could still be informative (*Dhawale et al., 2010*; *Adam et al., 2019*).

## Discussion

In this study we demonstrated and validated 3D-MAP, a one-photon technique that enables 3D multi-site random access illumination for high precision optogenetic photostimulation and imaging. 3D-MAP combines novel computational and optical advances to offer powerful and versatile optical brain manipulation capabilities. Unlike prior one-photon approaches that have low spatial resolution (DMD-based 2D projection) (*Dhawale et al., 2010*; *Leifer et al., 2011*; *Adam et al., 2019*; *Werley et al., 2017*; *Sakai et al., 2013*) or small FOV (SLM-based 3D CGH) (*Lutz et al., 2008*; *Anselmi et al., 2011*; *Szabo et al., 2014*; *Reutsky-Gefen et al., 2013*), 3D-MAP achieves high spatial precision at high speeds and in large 3D volumes, successfully addressing the superficial layers of neurons in the intact brain. 3D-MAP is the first system to achieve 3D multi-site illumination with previously unattainable DoF by simultaneously sculpting the light field in both intensity and angular domains. Therefore, 3D-MAP is able to generate high-resolution 3D patterns over large volumes that CGH cannot synthesize. 3D-MAP is also able to project 3D patterns at much faster speeds than CGH, not only because the current refresh rate of DMDs is an order of magnitude faster than that of SLMs, but also because calculating the light field parameters with ray tracing in 3D-MAP is much faster than calculating phase masks in 3D CGH. Perhaps most importantly for the broader neuroscience community, one can build a 3D-MAP system for a small fraction of the cost of two-photon holographic optogenetic systems (see *Appendix 1—table 2*). At the expense of lower spatial resolution and depth penetration, 3D-MAP is far simpler to operate and maintain and can readily address just as many neurons with far lower laser power. Since many types of circuit investigation experiments do not require the absolute spatial specificity of two-photon approaches (*Dhawale et al., 2010*; *Adam et al., 2019*), many groups that would not otherwise leverage patterned illumination approaches could adopt 3D-MAP for their optogenetics research applications.

3D-MAP is the first demonstration that uses 4D light field patterning for in vivo optogenetic photostimulation. Even though two-photon photostimulation techniques have combined SLMs with scanning mirrors previously (*Marshel et al., 2019*; *Packer et al., 2015*; *Yang et al., 2018*), 3D-MAP is fundamentally different from these techniques because it modulates the 4D light field $(x,\ y,\ k_x,\ k_y)$, so the DoF of the system is the *product* of the DMD's and the scanning mirrors' DoF. In previous work, the SLM and scanning mirrors both modulate 2D phase at the same plane $(k_x,\ k_y)$, so the DoF of the previous systems (*Marshel et al., 2019*; *Packer et al., 2015*; *Yang et al., 2018*) is the *sum* of the SLM's and the scanning mirrors' DoF. Hence, 3D-MAP achieves orders of magnitude more DoF without extra hardware. A further advantage of 3D-MAP over previous work is that it synthesizes a custom 3D intensity pattern in 3D $(x, y, z)$ with 4D light field control, which is a well-posed problem, whereas controlling 3D intensity with 2D phase control is an ill-posed problem. Note that previous work (*Levoy et al., 2009*) on 4D light field display also used a DMD, along with a microlens array (MLA), but is fundamentally different from 3D-MAP. First, the DoF of 3D-MAP is orders of magnitude higher since scanning mirrors have much more DoF than a fixed MLA. Second, the MLA has a built-in tradeoff between spatial and angular resolution whereas 3D-MAP does not. Therefore, the maximum defocus range of 3D-MAP is 720 µm (*Figure 2—figure supplement 1A*) whereas the MLA-based method only achieves 40 µm range in microscopy applications. Third, we demonstrated 3D-MAP with acute brain slices and in vivo mouse brain where tissue scattering distorts the light field and limits the resolution, whereas the previous work is not used for optogenetics under scattering.

As a proof-of-principle experiment, we demonstrated all-optical 3D-MAP in a relatively thin 3D volume (600 × 600 × 40 µm³) where two opsin-expressing neurons right on top of each other along the z-axis is very rare under the conditions of relatively sparse labeling (*Adesnik and Scanziani, 2010*). Also, because we used a patterned widefield imaging system to detect the fluorescence emitted from calcium indicators, the signal-to-noise ratio of the fluorescence images will be lower than the readout

noise of the camera if the neuron is too far away from the focal plane. To enable 3D detection over a larger axial range, one could apply pupil encoding (*Xue et al., 2019*), light field detection (*Prevedel et al., 2014*; *Pégard et al., 2016*), or remote focusing (*Botcherby et al., 2008*) to the imaging path of 3D-MAP.

Both one-photon and two-photon multi-site photostimulation are subject to spatial cross-talk, as out-of-focus light from one target may accidentally stimulate neurons that are at other focal planes, especially when the density of foci is high or when out-of-focus neurons are more photosensitive than the desired targets. In *Figure 7*, we demonstrated that the cross-talk in our experiment is minimal under conditions of relatively sparse opsin expression as obtained with conventional in utero electroporation (*Adesnik and Scanziani, 2010*). However, when the separation between two adjacent neurons is closer than the all-optical PPSF, off-target effects are likely, even under sparse expression conditions. Therefore, in addition to expression sparsity, we can also control the stimulation pattern's sparsity to mitigate spatial cross-talk. When the desired density of foci is high and cross-talk is expected, 3D-MAP could potentially leverage its multi-kHz patterning speed to exploit the temporal domain by multiplexing patterns and targeting smaller subsets of neurons at a time in order to increase pattern sparsity.

The maximum number of neurons that can be co-stimulated without cross-talk in one-photon or two-photon photostimulation depends on the available laser power, the limits of brain heating, the spatial resolution of the system, the FOV, and the spacing of opsin-expressing neurons. Compared to two-photon photostimulation, 3D-MAP trades on spatial resolution (by a factor of 2–3 in superficial layers) to use two to three orders lower laser power. For example, two-photon photostimulation techniques typically restrict the instantaneous power under the objective below 4 W to avoid photodamage (*Sridharan, 2021*; *Mardinly et al., 2018*). At full power, two-photon photostimulation techniques can simultaneously activate tens to over a hundred neurons (*Sridharan, 2021*; *Marshel et al., 2019*; *Mardinly et al., 2018*; *Dalgleish et al., 2020*). In contrast, 3D-MAP is able to photostimulate dozens of neurons simultaneously with only 480 µW out of the objective (*Figure 7F–G*). 3D-MAP requires sparser expression than for two-photon stimulation to achieve comparable effective specificity, but since total needed power is nearly 100-fold less for comparable stimulation patterns, brain heating is much less of a concern, and the technique can be scaled up to address more neurons in a larger brain area. Many experiments do not need full optical control over all neurons in a volume – for example, when targeting a specific transcriptional subtype, and many other experiments do not absolutely require single-cell specificity. Thus, for many applications 3D-MAP should be preferable to two-photon applications, with the specific exceptions of experiments that demand photostimulation of precise ensembles within a small volume of densely labeled tissue, or when the neural targets are located at brain depths only accessible to infrared light. The simplicity and scalability of 3D-MAP thus provides new experimental capabilities that two-photon photostimulation techniques cannot easily achieve.

Like any other one-photon photostimulation technique, the effective resolution of 3D-MAP in brain tissue is limited by scattering. As we show in *Figures 2 and 6*, illumination with red shifted sources can reduce Rayleigh scattering and mitigate resolution loss through brain tissue, but the accessible depth remains fundamentally limited to the first few hundreds of microns below the surface of the mouse brain and the scattering effect is more severe the deeper the stimulation is. To photostimulate neurons deeper inside the mammalian brain with 3D-MAP, the cortex can be surgically removed (*Adam et al., 2019*; *Dombeck et al., 2010*; *Marshel et al., 2012*) or implemented with a miniaturized microscope (*Szabo et al., 2014*; *Stamatakis et al., 2018*) as it is routinely performed to image deep structures such as the hippocampus or the thalamus. Taken together, 3D-MAP is a new volumetric optogenetic projection system that offers advantages over existing one-photon and two-photon optogenetic technologies and should facilitate a wide range of neural perturbation experiments to map the structure and function of brain circuits.

## Materials and methods
### 3D-MAP optical setup
The laser sources for optogenetic stimulation are diode pumped solid state (DPSS) laser diodes. One is at 473 nm wavelength (MBL-N-473A-1W, Ultralasers, Inc, Canada) and the other at 635 nm wavelength

(SDL-635-LM-5000T, Shanghai Dream Lasers Technology Co., Ltd., China) for different opsins. The results shown in *Figure 5* and *Figure 3—figure supplement 1E-H* are measured under red stimulation, and all the others are stimulated by blue light. The laser source for in vivo PPSF (*Figure 3H–J*), the widefield imaging (*Figure 4B*), and calcium imaging (*Figure 7*, *Figure 7—figure supplements 1 and 3*) is a DPSS laser at 589 nm wavelength (MGL-W-589–1W, Ultralasers, Inc, Canada). Current supplies are externally driven by an analog modulation voltage. The laser beams are scanned by a pair of galvo-mirrors (GVS202, Thorlabs, Newton, N.J.,U.S.), and then the beam size is expanded to fill the DMD (DLP9000X and the controller V4390, ViALUX, Germany) by a 4f system ($f_1 = 45$ mm, $f_2 = 150$ mm). The DMD is mounted on a rotation base in order to maximize the output laser power from 'ON' pixels and minimize the diffraction pattern from DMD pitches. Then the patterned beam passes through a tube lens ($f = 180$ mm) and the objective lens (XLUMPlanFL N, 20×, NA 1.0, Olympus) to generate multiple foci. The objective lens is mounted on a motorized z-stage (FG-BOBZ-M, Sutter Instrument, Novato, C.A., U.S.). A custom dichroic mirror (zt473/589/635rpc-UF2, Chroma, Bellows Falls, V.T., U.S.) is placed before the objective lens to reflect the stimulation laser beams while transmitting fluorescence photons emitted from the sample. The fluorescence passes through a tube lens ($f = 180$ mm) and a 4f system ($f_1 = 100$ mm, $f_2 = 150$ mm), and then is imaged by a camera (Prime95B, Teledyne Photometrix, Tucson, A.Z., U.S., *Figures 4B and 7*). Brain samples (acute brain slices and anesthetized mice) are placed on a motorized x-y stage (X040528 and the controller MP-285, Sutter Instrument, Novato, C.A., U.S.). The 3D PSF and patterns (*Figure 2* and *Figure 2—figure supplement 1*) are measured by capturing the fluorescence excitation in a thin fluorescent film on a microscope slide, with a sub-stage objective (XLUMPlanFL N, 20×, NA 1.0, Olympus) coupled to a camera (DCC1545M, Thorlabs, Newton, N.J., U.S.). Tomographic renderings of the 3D-MAP illumination patterns are obtained by mechanically scanning the illumination objective along the optical (z) axis and recording the 2D fluorescence image stacks at linearly spaced depths with the sub-stage camera. For *Figures 3A–D , and 4D–E*, the targeted positions in the 3D pattern are stimulated in a random order: two sequential stimulations are separated by a minimum distance calculated by Poisson disc sampling in order to avoid photocurrent accumulation caused by repeat stimulations. An NI-DAQ (National Instruments, NI PCIe-6363) synthesizes custom analog signals to synchronously modulate the lasers, the galvo-mirrors, the digital triggers to flip frames on the DMD, the trigger signals to the camera as well as the z-stage. An analog input channel enabling synchronous measurements of neural photocurrents and spikes in direct response to the custom 3D light sculpting sequence. A custom MATLAB (MathWorks, Natick, M.A., U.S.) graphic user interface is used to control the NI DAQ boards, calibrate and align the photostimulation and imaging modalities, and for data acquisition and processing.

## Synchronized control of 3D-MAP

The key mechanism to generate a focus on the shifted focal plane by 3D-MAP is to synchronize the scanning mirrors that control the angle of light ($\phi_{SM}$, $\theta$) (*Figure 8A*) and the DMD that controls the amplitude apertures. *Figure 8A* shows an example of synthesizing a focus below the native focal plane with 10 beams in 4 ms. The color of circular apertures and lines indicates the time stamp of the DMD apertures and corresponding projection angles. To better show the 3D view in *Figure 8A*, we plot the positions of DMD apertures and the projection angles in a 2D view (*Figure 8B–C*). The first aperture is located at $\phi_{DMD} = 0$ (dark blue circle, *Figure 8A–B*), and the corresponding projection angle is $\phi_{SM} = \pi$ (dark blue lines, *Figure 8A and C*). To project the angle $\phi_{SM} = \pi$, we apply –2 V voltage to the x-axis scanning mirror and 0 V to the y-axis scanning mirror (*Figure 8C–D*). Similarly, we can easily compute the other projection angles and the aperture positions by evenly dividing $2\pi$ by the number of beams. The voltage outputs to control the laser intensity, trigger DMD projection, and projection angles of the scanning mirrors are shown in *Figure 8D*. DMD projects the 10 apertures sequentially (*Figure 8E*) controlled by the TTL trigger signal (purple line, *Figure 8D*), while the scanning mirrors project the corresponding angles controlled by the sinusoidal signal (red line, voltage of x-axis scanning mirror; yellow line, voltage of the y-axis scanning mirror). Strictly, the scanning mirrors voltage should not change during the projection of one beam. However, because the time is very short (0.3 ms in *Figure 8D*), the change of projection angle during this time is negligible. Therefore, synchronizing the DMD and the scanning mirrors is straightforward and only involves calculating the voltage outputs and the DMD patterns as shown in *Figure 8D–E*.

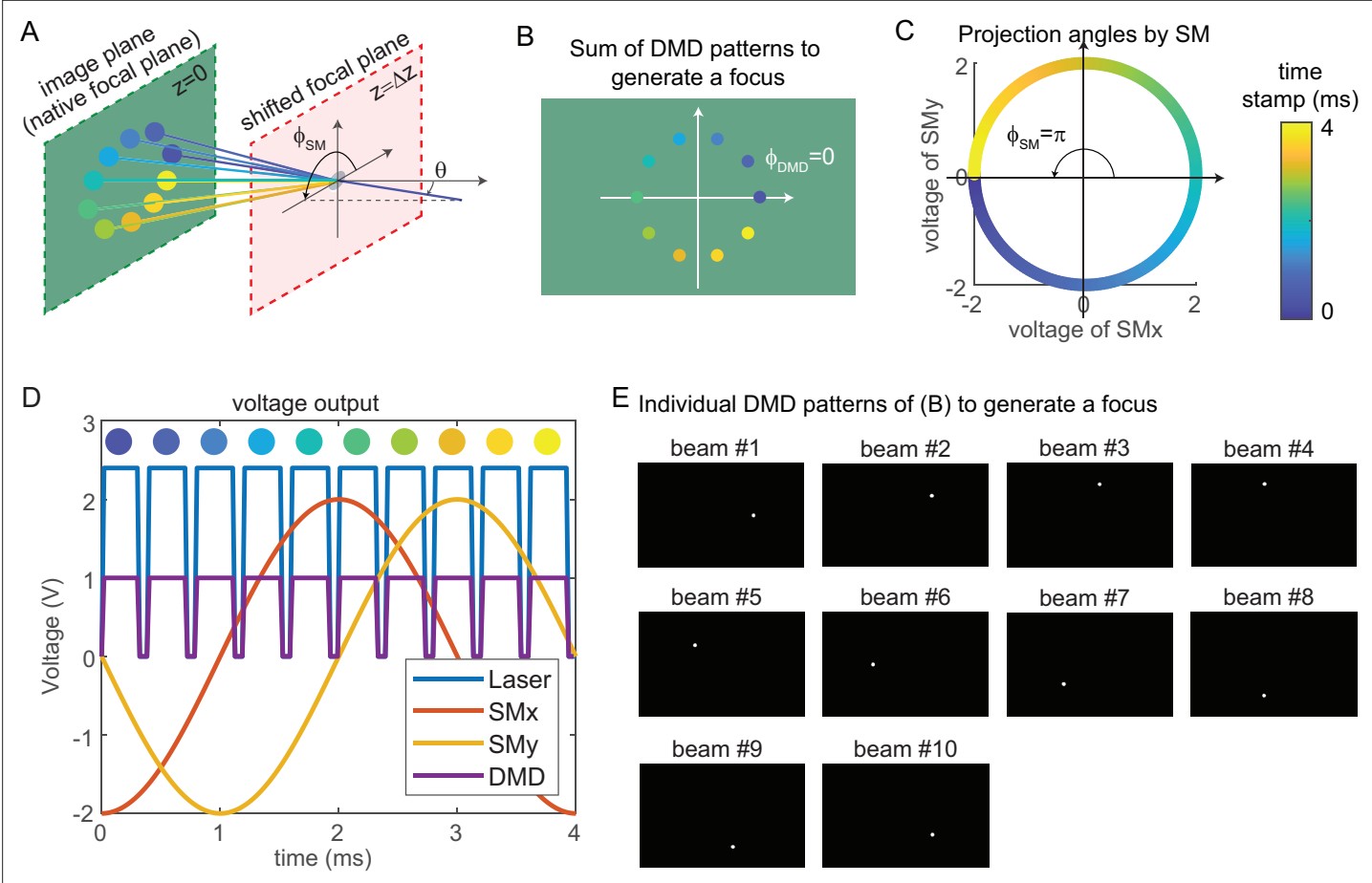

**Figure 8.** Synchronized control of the scanning mirrors and the digital micromirror device (DMD) to generate a focus at a shifted focal plane by three-dimensional multi-site random access photostimulation (3D-MAP). (**A**) A focus spot is generated by 10 beams in 4 ms in total. The DMD is located at the relayed image plane projecting 10 apertures sequentially. Scanning mirrors control the corresponding projection angles $(\phi_{SM}, \theta)$ for each beam. $\theta$ is the same for all beams and $\phi_{SM}$ varies. The color of the apertures and the projection angles shows their respective time stamp. (**B**) 2D view of the image plane in A, showing all 10 apertures on the DMD that are used to generate the focus. The first aperture (dark blue circle) is at $\phi_{DMD} = 0$ and the 10 apertures are evenly distributed in the range of $\phi_{DMD} = (0, 2\pi)$. (**C**) Projection angles by scanning mirrors. The first projection angle to generate the focus in A with the first aperture in B is located at $\phi_{SM} = \pi$. The scanning mirrors evenly scan along a circular trace in the 4 ms stimulation time. (**D**) The voltage outputs control the hardware. The scanning trace in C is generated by applying sinusoidal signals (red, yellow) to the scanning mirrors. The maximum voltage of the sinusoidal signal decides $\theta$ and the phase of the sinusoidal signal decides $\phi_{SM}$. The DMD projection is controlled by the TTL signal (purple), which has 10 rising edges in the 4 ms stimulation time to project 10 patterns sequentially. The laser intensity is controlled by an analog signal (blue) that is synchronized with the DMD. (**E**) The 10 patterns to be projected by the DMD to synthesize the focused spot.

Simulation results for typical misaligned examples and correct examples are shown in *Figure 9*. When the phase of DMD and the phase of the scanning mirrors are misaligned (*Figure 9A*), 3D-MAP cannot generate a tight focus at the shifted focal plane. Instead, the shape at the shifted focal plane looks like a ring (a paraboloid in 3D). To correct the phase mismatch, we can simply adjust $\phi_{DMD}$ or $\phi_{SM}$ to change the relative phase between the DMD patterns and the projection angles and correct for any system misalignment. For example, we can change $\phi_{SM}$ from $\frac{4\pi}{5}$ to $\pi$, as a result, the shape at the shifted focal plane will change from a ring to a spot (*Figure 9C*). Another misalignment may happen when the scanning mirrors have asymmetric response (*Figure 9B*), which generates an elliptical scanning trace rather than a circular trace. This type of misalignment degrades the focus of 3D-MAP exactly like optical astigmatism. To correct the asymmetric response of the scanning mirrors, we can adjust the amplitude of the voltage output to the scanning mirrors until we obtain a tight focus as in *Figure 9C*. In addition, adding a $\pi$-phase shift to the DMD patterns or the projection angles can generate a focus below (*Figure 9C*) or above (*Figure 9D*) the native focal plane.

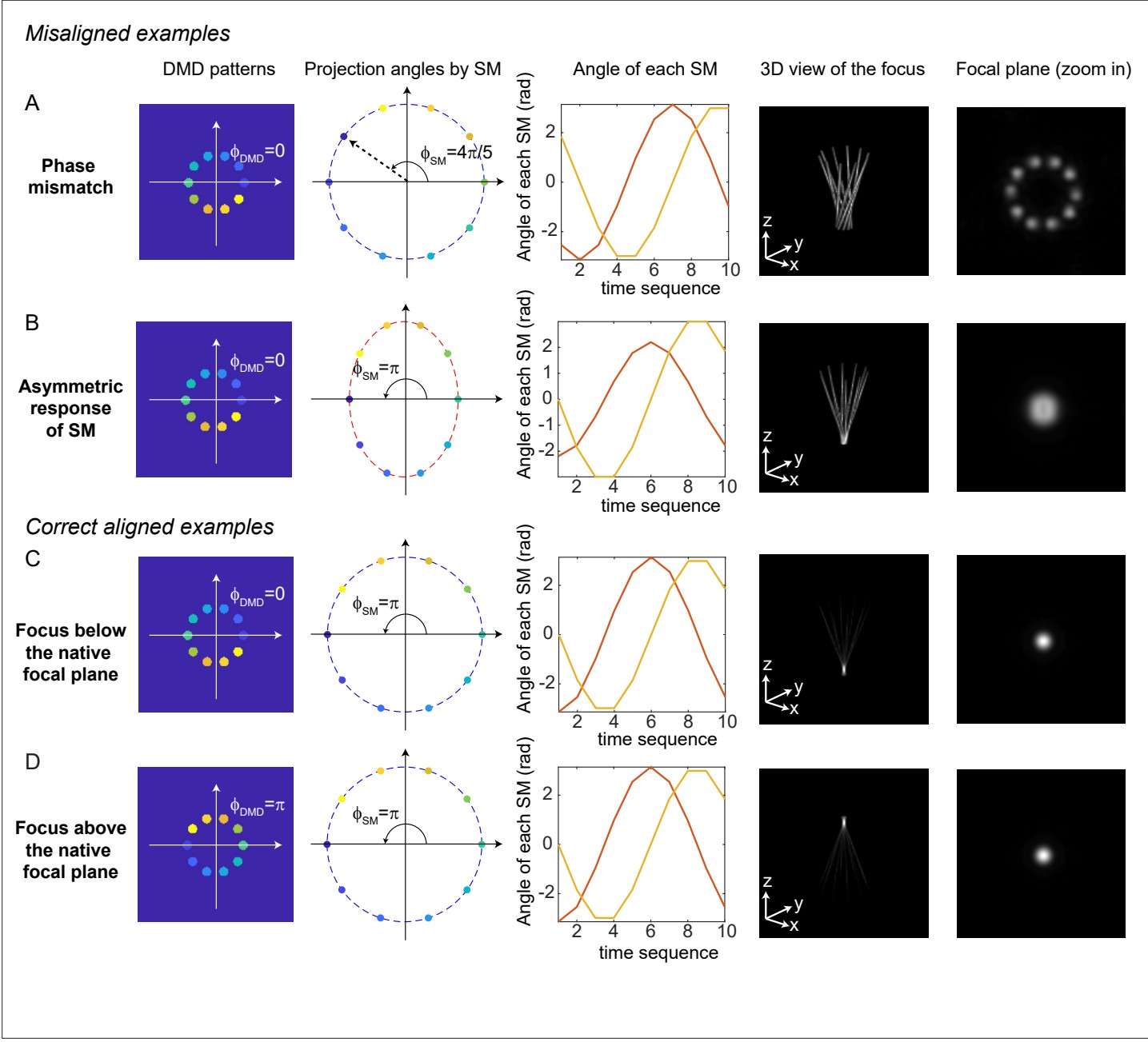

**Figure 9.** Misaligned examples (**A, B**) and correct aligned examples (**C, D**) with three-dimensional multi-site random access photostimulation (3D-MAP). The first column shows the sum of 10 digital micromirror device (DMD) patterns to generate the focus. The second column shows the projection angles by the scanning mirrors (SM). The color labels the timestamp of the projection as in *Figure 7*. The third column shows the angle of the scanning mirror in the x-axis (red) and the angle of the scanning mirror in the y-axis (yellow) for the sequential 10 projections, respectively. The fourth column shows the 3D view of the focus (distorted or tightly focused). The fifth column shows the 2D XY-view at the in-focus plane. To show the focus better, the images are zoomed in compared to the images in the fourth column. (A) Misaligned example of phase mismatching. The 10 beams cannot overlap to generate a tight focus, instead, a ring is generated at the shifted focal plane. (**B**) Misaligned example of asymmetric response of the scanning mirrors. The beams in the tangential direction and in the sagittal direction overlap at different locations along the z-axis like astigmatism, generating a distorted focus. (**C**) Correct aligned example generates a focus below the native focal plane (same as *Figure 7*). (**D**) Correct aligned example generates a focus above the native focal plane by adding a $\pi$-phase shift to the DMD patterns.

## Computational reconstruction framework

3D-MAP projects $N$ foci simultaneously in a 3D volume ($V$ voxels), which is defined as a stimulation pattern ( th pattern $A_i$). The positions of the $N$ foci are randomly selected from the $V$ voxels. The electrophysiology readout (EPSCs or IPSCs) under this pattern is recorded and the maximum absolute value of the EPSCs or IPSCs is the measurement $y_i$ under the illumination of  th pattern. Thus, the forward model of multi-site random simultaneous illumination is (**Figure 5A**):

$$y_i = \sum_{j=1}^{V} A_{i,j} x_j, \; i = 1, \; 2, \; ..., \; P.$$

$P$ is the number of patterns, and all patterns are orthogonal to each other. To solve the synaptic connection map at the $j$th voxel ($x_j$), the number $P$ should be equal to $V$. The reconstruction framework can be formulated as a $l_1$-regularized optimization problem that seeks to estimate $x$ by minimizing the difference between the measured currents ($y$) and those expected via the forward model (**Figure 5B**):

$$\underset{x \geq 0}{\mathrm{argmin}} \left( \|Ax - y\|_2^2 - \lambda R\left(x\right) \right),$$

where $R\left(x\right)$ describes total-variation (TV) regularization defined as:

$$R\left(x\right) = \sum_{j=1}^{V} \left| x_{j+1} - x_j \right|$$

This optimization problem is solved using fast iterative shrinkage-thresholding algorithm (FISTA), which is a first-order gradient descent algorithm. FISTA is able to reconstruct the result (**Figure 6C–D**, **Figure 6—figure supplement 1**) in real time during the experiments. The algorithm is summarized in Algorithm 1.

Algorithm 1. Three-dimensional multi-site random access photostimulation (3D-MAP) algorithm.

1. **Procedure** 3D-MAP multi-sites mapping reconstruction
2. Initialize $x_0$ by uniformly distributed random numbers between [0 1]
3. $k \leftarrow 0$
4. **while** $k < maxiter$ **do**
   a. Gradient $\Delta x \leftarrow FISTA \left[ x_0, A, y, \lambda \right]$
   b. $x_{k+1} = x_k - \mu \Delta x_k$, where μ is the step size
   c. $k = k + 1$
5. **return** $x$

If every voxel is illuminated $M$ times (the number of repetitions) using $P$ patterns and each pattern illuminates $N$ voxels, we can draw the relation between these parameters:

$$P = MV/N.$$

To solve $x$, $P$ should be equal to $V$, that is, the number of repetitions ($M$) should equal to the number of foci ($N$) in each pattern. However, the number of repetitions could be smaller than $M$ if the multiple illumination foci satisfy these two assumptions: first, the foci are distributed sparsely in the volume; second, the readout postsynaptic current is a linear combination of the response from presynaptic neurons stimulated by these foci. As shown in **Figure 6—figure supplement 1**, where the number of foci is five ($N = 5$), it is possible to reconstruct the synaptic connection map coarsely with less than five repetitions ($M = 1$–4) with compressive sensing algorithms.

## Animal preparation and electrophysiology

Neonatal mice age P3–P4 (wild type, *Emx1*-IRES-Cre [JAX stock#005628], or *Emx1*-Cre;*Gad1*-GFP [MGI:3590301]) were cryoanesthetized on ice and mounted in a head mold. AAVs driving Cre-dependent expression of either soma-targeted ChroME (**Figure 4**), Chrimson (**Figure 6**), ChRmine (**Figure 3—figure supplement 1**), or soma-targeted CoChR (**Figure 7** and **Figure 7—figure supplements 1–2**) were injected via a Nanoject3 (Drummond) into the visual cortex (~1–2 mm lateral to lambda, 3 sites, 22 nL/ site), ~100–300 μm below the brain surface. In wild type mice we injected AAV-mDlx-ChroME to drive opsin expression in cortical interneurons. All expression vectors co-expressed

nuclear-targeted mRuby3 except those in the all-optical interrogation experiments (*Figure 7* and *Figure 7—figure supplements 1–2*). Mice were used for brain slice or in vivo recordings at P28–P56. Brain slices were prepared as previously described (*Pluta et al., 2015*). Slices were transferred to a chamber and perfused with ACSF (no receptor blockers) warmed to ~33°C. First the microscope objective was centered over the area of V1 with the highest expression density of the opsin, as indicated by the density of mRuby3-expressing cells. ACSF contained in mM: NaCl 119, KCl 2.5, $MgSO_4$ 1.3, $NaH_2PO_4$ 1.3, glucose 20, $NaHCO_3$ 26, $CaCl_2$ 2.5. Internal solutions contained $CsMeSO_4$ (for voltage clamp) or KGluconate (for current clamp) 135 mM and NaCl 8 mM, HEPES 10 mM, $Na_3$GTP 0.3 mM, MgATP 4 mM, EGTA 0.3 mM, QX-314-Cl 5 mM (voltage clamp only), TEA-Cl 5 mM (voltage clamp only). For loose-patch experiments pipettes were filled with ACSF. The resistance of the patch electrodes ranged from 3 to 5 MΩ. For direct recording of photocurrents or light-induced spiking, we patched neurons (either in loose patch or whole-cell patch clamp) that expressed mRuby3. For recording light-driven synaptic inputs we patched from unlabeled putative interneurons (that did not have pyramidal morphology), or from GFP-expressing neurons in GAD67-GFP mice.

For optogenetic mapping we generated light spots ~10 μm wide (apertures of 10–20 pixels in radius on DMD). The total stimulation duration was 3–6 ms. For ChroME-expressing (*Figures 3A–E and 4*) and CoChR-expressing (*Figure 7* and *Figure 7—figure supplements 1 and 3*) animals, we used 473 nm light, and for Chrimson and ChRmine we used 635 nm light (*Figure 5*, *Figure 3— figure supplement 1*). All mapping used fully randomized sequences. For measurement of PPSFs and multi-spot synaptic maps, we used the system in full 3D mode (*Figures 3 and 5*, *Figure 4—figure supplement 1*). For single spot synaptic maps all photostimuli were at the native focal plane, and the microscope was moved mechanically under software control to obtain input maps at different axial planes (*Figure 4* and *Figure 4—figure supplement 1*). Mapping at the native focal plane requires only one DMD mask per light stimulus as compared to 10 for a full 3D pattern, allowing for many more masks to be stored in the DMD RAM and rapidly displayed. Scanning mirrors scan continuously in $2\pi$ to enable z-section when mapping at the native focal plane, which is different from direct 2D projection. At the beginning of each mapping experiment, we first took a 10-point laser power dose response curve of the photocurrent or light-induced synaptic current and used this data to choose the power range for photocurrent or synaptic mapping. For spiking PPSFs, the lowest light level that reliably generated spikes when directly targeting the soma was used. For single spot mapping we updated the DMD pattern at 40–80 Hz. For multi-spot mapping we updated the DMD pattern at 10–20 Hz, which minimized network adaptation.

For in vivo electrophysiology recording and all-optical interrogation, mice were sedated with chlorprothixene (0.075 mg) and anesthetized with isoflurane (1.5–2%). A small stainless steel plate was attached to the skull with Metabond. The skull was protected with cyanoacrylate glue and dental cement (Orthojet). A 2.5 mm craniotomy was made over V1 with a circular biopsy punch. The dura was removed with fine forceps and the craniotomy was covered with 1.2% agarose in ACSF and additional ACSF (electrophysiology) or covered with a cranial window (all-optical interrogation). The mouse was then injected with urethane for prolonged anesthesia (0.04 g), and supplemented with 0.5–1.5% isoflurane at the recording rig. Body temperature was maintained at 35–37°C with a warming blanket. Neurons were recorded under visualization either with epifluorescence (to target opsin-expressing neurons in the upper 100 μm of the brain for PPSF measurements) or in L2/3 via oblique infrared contrast imaging via an optic fiber (200 μm diameter) placed at a ~ 25 degree angle from horizontal located as close as possible to the brain surface underneath the objective. The same procedure for optogenetic mapping used in vitro was used in vivo. All data analysis was performed in MATLAB.

## Acknowledgements

This work was supported by DARPA N66001-17-C-40154 to HA and LW, as well as NIH UF1NS107574 to HA and LW. This work was supported by the New York Stem Cell Foundation. NP is a 2021 Beckman Young Investigator. HA is a New York Stem Cell Foundation Robertson Investigator. YX is a Weill Neurohub Fellow. This work was funded by the Gordon and Betty Moore Foundation's Data-Driven Discovery Initiative through Grant GBMF4562 to LW, and by the Burroughs Wellcome Fund, Career Award at the Scientific Interface (5113244) to NP. We thank Savitha Sridharan and Karthika Gopakumar for technical support.

## Additional information

### Funding

| Funder | Grant reference number | Author |
|---|---|---|
| Defense Advanced Research Projects Agency | N66001-17-C-40154 | Laura Waller<br>Hillel Adesnik |
| National Institutes of Health | UF1NS107574 | Laura Waller<br>Hillel Adesnik |
| New York Stem Cell Foundation | New York Stem Cell Foundation Robertson Investigator | Hillel Adesnik |
| Weill Neurohub | Fellowship | Yi Xue |
| Gordon and Betty Moore Foundation | GBMF4562 | Laura Waller |
| Burroughs Wellcome Fund | 5113244 | Nicolas Pégard |
| Arnold and Mabel Beckman Foundation | BYI 2021 | Nicolas Pégard |

The funders had no role in study design, data collection and interpretation, or the decision to submit the work for publication.

### Author contributions

Yi Xue, Conceptualization, Data curation, Funding acquisition, Methodology, Software, Validation, Visualization, Writing – original draft, Writing – review and editing; Laura Waller, Conceptualization, Funding acquisition, Supervision, Writing – review and editing; Hillel Adesnik, Data curation, Funding acquisition, Methodology, Resources, Supervision, Validation, Writing – review and editing, Conceptualization, Investigation, Project administration, Writing – original draft; Nicolas Pégard, Conceptualization, Funding acquisition, Methodology, Writing – review and editing

### Author ORCIDs

Yi Xue https://orcid.org/0000-0003-2622-083X
Laura Waller https://orcid.org/0000-0003-1243-2356
Hillel Adesnik https://orcid.org/0000-0002-3796-8643
Nicolas Pégard https://orcid.org/0000-0003-2868-7118

### Ethics

**Ethical statement.**All animal experiments were performed in accordance with the guidelines and regulations of the Animal Care and Use Committee of the University of California, Berkeley. Protocol ID: AUP-2014-10-6832-2.

### Decision letter and Author response

Decision letter https://doi.org/10.7554/eLife.73266.sa1
Author response https://doi.org/10.7554/eLife.73266.sa2

## Additional files

### Supplementary files

• Transparent reporting form

### Data availability

Data of electrophysiology measurement and optical imaging has been deposited in Github (https://github.com/Waller-Lab/3D-MAP). Custom code used to collect and process data is programed in MATLAB. The code has been deposited in Github (https://github.com/Waller-Lab/3D-MAP, copy archived at swh:1:rev:6b669d624e50bbf4984ccc3b93bb764ecc0548c6).

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

# Appendix 1

## Supplementary materials

**Appendix 1—table 1.** Comparison of light-targeting photostimulation methods.

| Specific technique | Two-photon optogenetics | One-photon optogenetics | | | Our technique 3D-MAP |
| --- | --- | --- | --- | --- | --- |
| | | Scanning methods (GM and AOD)* | DMD‡ projection | CGH § | |
| Pros | · High 3D resolution · High penetration depth · Less cross-talk because of non-linear excitation | • Low power illumination • Compact and inexpensive systems | | | |
| | | • Reduced cross-talk with sparse excitation | • Fast patterning speed • Large field-of-view | • High 3D resolution | • Large number of DoF • Fast patterning speed • High 3D resolution • Large field-of-view • Less cross-talk from out-of-focus light |
| Cons | • Moderate numbers of targets • Small accessible volume • Small numbers of degrees-of-freedom • High-power illumination (heat and photodamage) • Expensive and sophisticated systems • Slow patterning speed | • Rayleigh scattering limits penetration depth • Lower axial resolution than two-photon optogenetics | | | |
| | | • No simultaneous multiple stimulation • Small number of DoF‡ | • Low resolution • No depth specificity • 2D modulation • Small number of DoF† • Cross-talk from out-of-focus light | • Small number of targets • Small accessible volume • Small number of DoF • Slow patterning speed • Cross-talk from out-of-focus light | • Lossy amplitude modulation that requires bright laser sources |
| Refs | *Carrillo-Reid et al., 2019*; *Naka et al., 2019*; *Daie et al., 2021*; *Sridharan, 2021*; *Nikolenko et al., 2008*; *Papagiakoumou et al., 2010*; *Pégard et al., 2017* | *Robinson et al., 2020*; *Forli et al., 2021*; *Mardinly et al., 2018* | *Mardinly et al., 2018*; *Leifer et al., 2011*; *Adam et al., 2019*; *Werley et al., 2017*; *Sakai et al., 2013* | *Lutz et al., 2008*; *Anselmi et al., 2011*; *Szabo et al., 2014*; *Reutsky-Gefen et al., 2013* | This study |

*GM: galvo mirrors. AOD: acousto-optic deflectors.
†DoF: degrees of freedom.
‡DMD: digital micromirror device
§CGH: computer generated holography.

**Appendix 1—table 2.** Comparison of the cost of one-photon 3D-MAP and two-photon 3D-SHOT (*Pégard et al., 2017*; *Mardinly et al., 2018*).

| Description | 1P 3D-MAP | | | 2P 3D-SHOT | | |
| --- | --- | --- | --- | --- | --- | --- |
| | Part # | Budget | High performance | Part # | Budget | High performance |
| Photostimulation and imaging lasers | Blue DPSS laser | $700 | $9000 | Femtosecond laser for photostimulation | $80,000 | $160,000 |
| | Yellow DPSS laser | $3500 | $13,000 | Femtosecond laser for calcium imaging | $80,000 | $140,000 |
| Imaging sensor/ system | sCMOS camera | $5000 | $25,000 | PMTs and acquisition system | $20,000 | $40,000 |
| Light modulator | DMD and the controller | $2000 | $15,000 | LCoS-SLM | $15,000 | $60,000 |
| | Scanning mirrors | $2500 | $6400 | | | |
| Optical table | min 3'×3' | $4000 | $7500 | min 4'×6' | $8500 | $10,000 |
| Total | | $17,700 | $75,900 | | $203,500 | $410,000 |

For one-photon (1P) 3D-MAP, the budget system costs 23% of the high-performance system at an expense of performance. First, lasers with lower cost or even LEDs can photostimulate opsins, but the maximum power is usually much lower than that of expensive 1P lasers. 3D-MAP

distributes the total power from the laser across the whole FOV. To keep the same power density for efficient photostimulation, using a lower power laser will result in a smaller FOV for the same sample. Second, sCMOS cameras at lower price may have fewer pixels, higher readout noise, or slower frame rate, which will generate more noisy measurement of calcium signals. Third, a budget DMD has fewer pixels and slower pattern rate (but still above 1 kHz) than a high-performance DMD, and budget scanning mirrors have smaller size and scanning range. These factors will limit the total number of voxels, that is, the system will either have low resolution across a large accessible volume or high resolution across a small volume. Finally, the main factor affecting optical table cost will be the thickness of the table, given 3D-MAP is compact in lateral dimension.

**Appendix 1—table 3.** Summary of experiment details.

| Figure | Type | Sample | Label | DMD aperture size | Imaging parameters |
|---|---|---|---|---|---|
| *Figure 2* | Widefield imaging with a sub-stage camera | Uniform fluorescent calibration slide | Fluorescence paint from Tamiya Color, mouse brain slices are label-free | *Figure 2A–C*: 1 pixel *Figure 2D–J*: 10 pixels | Capture one widefield image at each z-plane. Imaging speed: 10 Hz |
| *Figure 3* | Photostimulation and electrophysiology | *Figure 3A–G*: brain slices *Figure 3H–J*: in vivo | L2/3 excitatory neurons expressing soma-targeted ChroME | *Figure 3A–D*: 15 pixels *Figure 3E–J*: 10 pixels | N/A |
| *Figure 4* | *Figure 4B*: widefield fluorescence imaging *Figure 4C–G*: Photostimulation and electrophysiology | Brain slices | L2/3 excitatory neurons expressing soma-targeted ChroME | *Figure 4C*: 50 pixels *Figure 4D–G*: 20 pixels | Captured one widefield image. Exposure time: 100 ms |
| *Figure 5* | Photostimulation and electrophysiology | Brain slices | L2/3 excitatory neurons expressing soma-targeted ChroME | *Figure 5C–D*: 20 pixels | N/A |
| *Figure 6* | Photostimulation and electrophysiology | Brain slices | L2/3 excitatory neurons expressing Chrimson | *Figure 6C–E*: 10 pixels | N/A |
| *Figure 7* | Photostimulation and calcium imaging | *Figure 7B*: brain slices *Figure 7C–I*: in vivo | L2/3 excitatory neurons co-expressing soma-target CoChR and red calcium indicator jRCaMP1a | *Figure 7B–I*: 10 pixels | Exposure time: 3.8 s Imaging speed: 10 Hz |

