## [Editor Report]

This paper is of interest to neuroscientists working on all-optical interrogation of neural circuits and optogenetics. It provides a new, inexpensive, one-photon approach for high-speed 3D photostimulation with sparse targeting. This new method has been well characterized and demonstrated in both in vitro and in vivo experiments on mouse brain tissue.

---

## [Decision Letter]

**Decision letter after peer review:**

Thank you for submitting your article "Three-dimensional Multi-site Random Access Photostimulation (3D-MAP)" for consideration by *eLife*. Your article has been reviewed by 3 peer reviewers, and the evaluation has been overseen by a Reviewing Editor and John Huguenard as the Senior Editor. The following individual involved in review of your submission has agreed to reveal their identity: Luis Alberto Carrillo Reid (Reviewer #3).

The reviewers have discussed their reviews with one another and are unanimous in recommending submitting a revised version of the manuscript. The Reviewing Editor has drafted this to help you prepare a revised submission.

Essential revisions:

All 3 reviewers were enthusiastic about the manuscript and found your approach of an all optical 1P method for stimulating neurons to be of interest to and impactful for the neuroscience field. In general, the reviewers found the manuscript to be comprehensive and well-written. However, there were several points of clarification/ more detailed explanations the reviewers have requested that will be need to addressed prior to recommending publication in *eLife*. Please see the detailed points of clarification requested by reviewers below.

*Reviewer #1:*

The authors nicely quantify the optical point spread function both on a fluorescent slide and, importantly, in brain slices and in vivo, and they do so for blue and red light – wavelengths that are suited for single photon optogenetics. The authors demonstrate the utility of this technique, particularly in cases where neurons expressing a given optogenetic protein are sparsely labeled or, as may be the case for inhibitory neurons, naturally sparse. Notably, as cell type specific promoters in layer 2/3 are more fully characterized, it is likely that naturally sparse excitatory networks will be evident as well.

The authors demonstrate the use of this approach for optogenetic circuit mapping, and for in vivo read-write experiments.

Weaknesses:

In Figure 4E, it seems like excitatory inputs to PV cells are clustered. Is this the correct message from this figure, or do those spots (particularly near dZ=40um) represent single excitatory neurons rather than groups of excitatory neurons?

I didn't understand the approach in Figure 6. CoChR is a blue-light opsin while jRCaMP1a is a red light calcium indicator. Why the need to start imaging the jRCaMP1a responses only after the CoChR stimulation? Is there a great deal of cross talk in the excitation wavelengths?

While the authors are very up front about the fact that this stimulation approach works best for very sparsely labeled neurons, this does limit the potential utility of the application.

Overall, this is a well written paper that presents a full accounting of a new and potentially quite useful 3D stimulation approach. The authors are candid about the shortcomings, which is refreshing. They provide well done quantitative measures and they do so across a range of preparations. They validate their approach with electrophysiological measures. There is little here to dislike.

I have no specific recommendations. I found the paper to be well written and I imagine that it will be of some real interest to the field.

*Reviewer #2:*

Xue et al., introduced a new high-speed 3D photostimulation method, 3D-MAP, based on single photon light sculpting. Compared with typical single photon approaches, this new technique is improved in 3D spatial resolution, simultaneous targeting numbers and pattern refresh rate, offering a simple and inexpensive alternative to two-photon optogenetics. Although this technique requires a certain level of expression sparsity especially in a scattering sample, its overall 3D stimulating throughput still makes it valuable in various applications.

Xue et al., also demonstrated the application of 3D-MAP in in vivo 3D optogenetic photostimulation and all-optical parallel interrogation of mouse brain with impressive results. The 3D-MAP technique can be potentially used as an add-on module for existing imaging system, such as for single objective light-sheet microscopy. Its application can be expanded to other animal models with a less scattering brain, such as zebrafish, to mitigate spatial crosstalk.

This paper has provided sufficient technical details for readers to understand the 3D-MAP method, with carefully designed and conducted experiments to demonstrate the potential applications in high-throughput interrogation of neural circuits. Yet there are some aspects of the system characterizations and experiment details that need to be further clarified.

Overall, this manuscript described the 3D-MAP technique clearly and demonstrated its application with promising data. I do have a few suggestions that I think would enhance the clarity and strength of this paper.

1. The authors mentioned that they could unselect certain illumination angels of specific stimulated focus to avoid stimulating non-targeted areas. Is it calculated automatically with the current algorithm? When multiple rays for different foci make a cross section in the 3D space, do they get removed or not?

2. The authors compared different foci across the large FOV in Figure 2—figure supplement 1 based on the 25 foci located on an oblique plane (Figure 2D). It would be more reliable if the PSF is averaged from multiple measurements taken from randomly distributed foci across the entire FOV.

3. Is the same 10-pixel radius aperture on the DMD used for all the experiment with mouse brain tissue later? Is the PSF improved in scattering tissue when using a smaller radius?

4. The authors performed the all-optical interrogation experiment in vivo in L2/3 neurons that are 200μm-300μm deep in the intact mouse brain, yet they only measured the optical PSF under the brain slice up to 150μm. Similarly, the PPSF measurement in Figure 3 only shows the results "in the upper 100μm of the brain". Since the resolution in Z degraded dramatically with blue light stimulation, it would be better to add the measurement of thicker brain tissue to mimic the in vivo experiment condition in L2/3. Besides, the all optical 3D PPSF under 200μm (Figure 6E) shows a similar axial FWHM compared with the electrophysiology PPSF in the upper 100μm brain (Figure 3I) as well as the optical PSF in under 100μm brain slice (Figure 2H). This is a bit confusing and could be a bit misleading. Please explain further taking the optical PSF, the nonlinear effect of opsin and calcium indicator sensitivity and the imaging system performance into consideration.

*Reviewer #3:*

This paper proposes a low-cost approach to perform simultaneous calcium imaging recordings and photostimulation of neuronal ensembles. The study of the causal relation between the activity of identified groups of neurons and their behavioral output is a topic of broad interest for the neuroscience community. Recently it has been shown in different brain areas that the identification and further manipulation of neuronal ensembles related to a behavioral task could be achieved using two-photon microscopy. This paper proposes a one photon tool to eventually facilitate the use of all-optical interrogation of neuronal circuits and behavior. However, the main limitations of the system showed in this paper are the necessity of sparse expression of opsins and the spatial resolution that comprises volumes bigger than the somatic region of individual neurons.

The authors suggest that the proposed 1P system could be used instead of 2P systems but according to the experimental results, this is only valid for a very narrow set of experiments that involve the sparse expression of opsins.

1. To clearly show the ability of the system to perform simultaneous calcium imaging and photostimulation the authors should show raw calcium transients of photostimulated and non-photostimulated neurons at different trials, as well as a plot showing calcium transients of non-stimulated neurons that express the opsin as a function of the distance between photostimulated neurons.

2. The manuscript mentioned the cost of budget and high performance realizations of the 1P system but a clear statement of the disadvantages of a budget system is lacking. It would be helpful to include a paragraph indicating what is missing in a budget 1P system compared to a high performance 1P system.

3. One of the main issues with all-optical interrogation of neural circuits and behavior is the lack of co-expression of opsins and calcium indicators. This is a crucial step since usually the targeted neurons are the ones that have activity in the behavioral task. However, the manuscript mentions that the system proposed is ideal for sparse expression of opsins, but the authors didn't mention the level of co-expression in their experiments. If the level of co-expression in a sparse labeling is low, then the system will not be ideal for the all-optical interrogation of neural circuits and behavior.

4. The main strength of the paper is the demonstration that the system could be used to map the influence of sparse neurons over a large volume onto a postsynaptic target. So, focusing the rationale, results and conclusions on those experiments could be a better strategy until the 1P system is refined to reach similar performance of a 2P system for broad expression of opsins.

---

## [Author Response]

Reviewer #1:The authors nicely quantify the optical point spread function both on a fluorescent slide and, importantly, in brain slices and in vivo, and they do so for blue and red light – wavelengths that are suited for single photon optogenetics. The authors demonstrate the utility of this technique, particularly in cases where neurons expressing a given optogenetic protein are sparsely labeled or, as may be the case for inhibitory neurons, naturally sparse. Notably, as cell type specific promoters in layer 2/3 are more fully characterized, it is likely that naturally sparse excitatory networks will be evident as well.The authors demonstrate the use of this approach for optogenetic circuit mapping, and for in vivo read-write experiments.Weaknesses:In Figure 4E, it seems like excitatory inputs to PV cells are clustered. Is this the correct message from this figure, or do those spots (particularly near dZ=40um) represent single excitatory neurons rather than groups of excitatory neurons?

We apologize for the lack of clarity. In Figure 4E and the corresponding traces in Figure 4F, the size of each cluster is about 30µm (2-4 pixels, 9µm/pixel), which reaches the limit of the spatial resolution shown in Figure 3A-B (lateral PPSF 29±0.8µm). therefore, we cannot distinguish whether the spatial cluster of candidate presynaptic input contains just a single presynaptic neuron. However, the fact that most of the postsynaptic responses from this cluster had very similar amplitudes and time courses suggest (but does not prove) that most of the responses were indeed due to a single primary source. We added this explanation in the revised manuscript (line 326-329) to avoid confusion.

I didn't understand the approach in Figure 6. CoChR is a blue-light opsin while jRCaMP1a is a red light calcium indicator. Why the need to start imaging the jRCaMP1a responses only after the CoChR stimulation? Is there a great deal of cross talk in the excitation wavelengths?

We apologize for the lack of clarity. Even though jRCaMP1a is a red light calcium indicator, we noticed the blue light at 473nm can stimulate jRCaMP and result in an increase of fluorescence intensity. Such rise of fluorescence intensity is not due to calcium activity but to cross talk with the excitation wavelengths (the 1P absorption spectrum of jRCaMP1a is reported in the Response ref.1). Therefore, we choose to start imaging the jRCaMP1a after the CoChR stimulation. With this carefully calibrated system, we ensure that any detected rise of fluorescence intensity can be attributed to calcium activity, and not to any other artifacts (see the control experiment in Figure 6B). to clarify this point, we added this explanation in the revised manuscript in line 471-475.

While the authors are very up front about the fact that this stimulation approach works best for very sparsely labeled neurons, this does limit the potential utility of the application.

We agree that the requirement of sparsity constrains the applications for our technology for precise ensemble stimulation. However, there are other applications that will not require sparse expression when absolute cellular specificity is not required. Numerous prior studies have used much coarser (non-3D) one-photon stimulation for mapping connectivity and for probing causal relationships between area and cell-type specific activity in behaviour. £d-MAP is preferable to these approached even with dense opsin expression because it still provides a significant leap in spatial resolution, but perhaps even more importantly, volumetric control.

With respect to precise ensemble stimulation, our manuscript mentions that 3D-MAP is not suitable for experiments that require photo-stimulation of precise ensembles within a small volume of densely labelled tissue, where 2Pphotostimulation is sufficient (line 652-655). Instead, our technology enables at the expense of a slightly lower spatial specificity the scalability that multiphoton techniques will never achieve because of fundamental tissue heating issues built into the low efficiency of nonlinear optical absorption. We indicate that loss of spatial specificity can be addressed either by leveraging sparser labelling (expressing the optogenetic modulator in a smaller number of randomly distributed neurons), or computationally, by targeting smaller patterns significantly faster that the response speed of opsins. We discuss the sparsity requirement in depth in the Discussion section.

Overall, this is a well written paper that presents a full accounting of a new and potentially quite useful 3D stimulation approach. The authors are candid about the shortcomings, which is refreshing. They provide well done quantitative measures and they do so across a range of preparations. They validate their approach with electrophysiological measures. There is little here to dislike.I have no specific recommendations. I found the paper to be well written and I imagine that it will be of some real interest to the field.

We thank the reviewer for the positive comments.

Reviewer #2:Xue et al., introduced a new high-speed 3D photostimulation method, 3D-MAP, based on single photon light sculpting. Compared with typical single photon approaches, this new technique is improved in 3D spatial resolution, simultaneous targeting numbers and pattern refresh rate, offering a simple and inexpensive alternative to two-photon optogenetics. Although this technique requires a certain level of expression sparsity especially in a scattering sample, its overall 3D stimulating throughput still makes it valuable in various applications.Xue et al., also demonstrated the application of 3D-MAP in in vivo 3D optogenetic photostimulation and all-optical parallel interrogation of mouse brain with impressive results. The 3D-MAP technique can be potentially used as an add-on module for existing imaging system, such as for single objective light-sheet microscopy. Its application can be expanded to other animal models with a less scattering brain, such as zebrafish, to mitigate spatial crosstalk.This paper has provided sufficient technical details for readers to understand the 3D-MAP method, with carefully designed and conducted experiments to demonstrate the potential applications in high-throughput interrogation of neural circuits. Yet there are some aspects of the system characterizations and experiment details that need to be further clarified.

We thank the reviewer for the positive feedback. We agree that 3D-MAP ha strong potential to become a popular add-on light sculpting module for existing imaging systems, and that it can be easily scaled and implemented for a broad range of animal models, in particular zebrafish where it will highly benefit from lower scattering. Please find below more details to answer the reviewers’ questions.

Overall, this manuscript described the 3D-MAP technique clearly and demonstrated its application with promising data. I do have a few suggestions that I think would enhance the clarity and strength of this paper.1. The authors mentioned that they could unselect certain illumination angels of specific stimulated focus to avoid stimulating non-targeted areas. Is it calculated automatically with the current algorithm? When multiple rays for different foci make a cross section in the 3D space, do they get removed or not?

We thank the reviewer for bringing point up. We revised the code to add the capability of unselecting certain illumination angles to avoid stimulating non-targeted areas (see “3D-MAP simulation demo” in our github: https://github.com/Waller-Lab/3D-MAP) and mentioned this point in the revised manuscript (line 159-161). We simulated the cases with and without a non-targeted areas shown in Figure 1—figure supplement 2. The simulation generates two foci of different size, one above the native focal plane and the other one is below the focal plane. When there is no non-targeted area near the centre, the foci are synthesized by 10 illumination angles (Figure 1—figure supplement 21A-C); with a non-targeted area near the centre, one illumination angle of the focus near the non-targeted area is removed (pointed by the yellow arrow, Figure 1—figure supplement 2).

We can either remove rays or alternate illumination angles to avoid stimulation non-targeted areas. Here, we choose to remove rays simply by closing the DMD aperture at the desired location for the frame matched to any particular incident wave direction. Our algorithm first computes a fully populated 3D illumination, with in e ray for each available illumination angle. Then, we check whether the light intensity in the non-targeted areas is above a user-defined threshold. If so, we identify the rays that pass through this area and their corresponding DMD apertures, we then modify the DMD patterns to remove these apertures. All the other apertures remain in place (Figure 1—figure supplement 2D).

2. The authors compared different foci across the large FOV in Figure 2—figure supplement 1 based on the 25 foci located on an oblique plane (Figure 2D). It would be more reliable if the PSF is averaged from multiple measurements taken from randomly distributed foci across the entire FOV.

We agree with the authors that the PSF in figure 2A-C shows the best optical spatial resolution 3D-MAP can achieve, which is measured by turning on a one-pixel aperture in the centre of the DMD (acting as a source point) and scanned through angles. The resolution of 3D-MAP will degrade when the focus is not paraxial. We added a statement in line 208-210 to clarify this point:

“This PSF is measured in the centre of the FOV and represents the best optical resolution of 3D-MAP, while the resolution will degrade when the focus is not paraxial.”

On the other hand, the 25 foci in Figure 2D and Figure 2—figure supplement 1 is merely chosen to show that we can generate multiple foci simultaneously in 3D, and allows us to quantitatively evaluate the change of the foci’s size in non-ideal conditions, when targets are away from the centre of the FOV.

3. Is the same 10-pixel radius aperture on the DMD used for all the experiment with mouse brain tissue later? Is the PSF improved in scattering tissue when using a smaller radius?

We apologize for the lack of clarity. We used different radius apertures for different experiments, but all of them are equal or larger than 10 pixels. We need to find a balance between laser power, resolution, and scanning speed when we determine the radius of aperture. Since the maximum power density is determined by the lasers, for less sensitive opsin or less expression level, we use a larger aperture to make sure the total power is high enough to evoke photo-activity. The amount of light on any given target is quadratically proportional to the radius of the aperture. For very sensitive opsins, we used smaller apertures to improve the resolution. For course scanning over a large FOV, such as Figure 4C, using a large aperture with fewer scanning steps can quickly scan the entire FOV. If we could use lasers with higher power and more sensitive opsins, we could further reduce the radius of aperture and achieve higher spatial resolution especially in scattering tissues. We added the aperture size to the new Table 3 in the revised manuscript. We also mention this information in line 227-228.

4. The authors performed the all-optical interrogation experiment in vivo in L2/3 neurons that are 200μm-300μm deep in the intact mouse brain, yet they only measured the optical PSF under the brain slice up to 150μm. Similarly, the PPSF measurement in Figure 3 only shows the results "in the upper 100μm of the brain". Since the resolution in Z degraded dramatically with blue light stimulation, it would be better to add the measurement of thicker brain tissue to mimic the in vivo experiment condition in L2/3. Besides, the all optical 3D PPSF under 200μm (Figure 6E) shows a similar axial FWHM compared with the electrophysiology PPSF in the upper 100μm brain (Figure 3I) as well as the optical PSF in under 100μm brain slice (Figure 2H). This is a bit confusing and could be a bit misleading. Please explain further taking the optical PSF, the nonlinear effect of opsin and calcium indicator sensitivity and the imaging system performance into consideration.

We apologize for the lack of clarity. We agree with the reviewer that the measured affective resolution (PPSF) will depend on multiple factors, and that the all-optical PPSF is complicated by sensitivity of the calcium sensor. Acquiring an electrophysical PPSP deeper than 100 microns in tissue is not feasible without 2p imaging. Since the 3D-MAP microscope only has wide-field (1p) imaging this was not possible, however we have added to the text that the electrophysiological PPSF in Figure 3A is a best-case scenario (line 290-292). The all-optical PPSF (PPSF_o_) in new Figure 7E may be similar to the ephys PPSF in Figure 3H-J, despite being 50-100 microns deeper, because jRCaMP1a probably cannot detect very low spike rates. We have also added this to the text in line 1276-1286.

More generally, the PSF is defined as the response of an imaging system to a point source. Here, we distinguish the optical PSF (PSF_o_), the physiological PSF (PPSF), and the all-optical PPSF (PPSF_o_) as impulse responses to three different systems (see figure 7—figure supplement 2). For example, the of PSF_o_ is quantified by measuring the full-width-half-maximum (FWHM), but the light intensity at half maximum may not be enough to evoke photo-induced neural activity. Therefore, the size of the “effective focus” seen by opsin molecules may be smaller than the size of PSF_o_. The PPSF is also related to the fraction of opsin molecules that must be optically gated to drive the neuron to spike, which in turn depends on the opsin conductance, the total opsin expression levels, and the intrinsic excitability of the neuron. This is because the closer one is to saturate the opsin molecules on a given neuron to generate a spike, the broader the PPSF will be. Thus the PPSF evaluates a different system than the optical excitation path alone, On the other hand, PPSF_o_ uses calcium indicators and optical detection path to measure neural activity, which evaluates the system consisting of optical excitation path, opsins, calcium indicators, and optical detection path. By comparing PSFs, one can understand how losses of spatial specificity are distributed between limitations of the optical system, optical scattering in tissue, uneven opsin distribution, and nonlinear biological response.

While we agree that adding the measurement of thicker brain tissue could provide more information of optical scattering, we are confident that the PSF_o_ measured through a thicker sample won’t provide enough information to mimic the in vivo experiment condition in L2/3. The purpose of Figure 2E-J is to show how PSF_o_ changes with different levels of scattering, rather than mimic in vivo experiment conditions. That is why we measure in vivo PPSF_o_ as another system characteristic when we perform in vivo experiments in L2/3 (new Figure 7E).

To clarify this point, we added to the revised manuscript figure 7—figure supplement 2, as well as this discussion in line 1276-1286.

Reviewer #3:This paper proposes a low-cost approach to perform simultaneous calcium imaging recordings and photostimulation of neuronal ensembles. The study of the causal relation between the activity of identified groups of neurons and their behavioral output is a topic of broad interest for the neuroscience community. Recently it has been shown in different brain areas that the identification and further manipulation of neuronal ensembles related to a behavioral task could be achieved using two-photon microscopy. This paper proposes a one photon tool to eventually facilitate the use of all-optical interrogation of neuronal circuits and behavior. However, the main limitations of the system showed in this paper are the necessity of sparse expression of opsins and the spatial resolution that comprises volumes bigger than the somatic region of individual neurons.The authors suggest that the proposed 1P system could be used instead of 2P systems but according to the experimental results, this is only valid for a very narrow set of experiments that involve the sparse expression of opsins.

We thank the reviewer for positive comments in the relevance of or work for experimental neurosciences. Here, we would like to address specifically the reviewers concern regarding the drawbacks of 3D-MAP in comparison with multiphoton systems. As we showed in Supplementary materials – table 1, it is well established that one-photon and two-photon photo-stimulation systems have their respective pros and cons. Here, our work on 3D-MAP aims to explore in-depth the untapped potential of one-photon technology. We agree that much like many one-photon systems, 3D-MAP has lower spatial resolution and requires sparser expression than for 2P photo-stimulation to achieve comparable effective specificity, however, these limitations must be put in perspective with the benefits of scalability. 3D-MAP only needs about 100-fold less laser power for comparable stimulation patterns, and brain heating is much less of a concern, so the technique can be scales up to address more neurons in a larger brain area (line 641-649). Or manuscript highlights how 3D-MAP can enable experiments in conditions that are not suitable for 2P applications, and as such, should be considered, not as a replacement for 2P methods, but as a complimentary approach with a rather orthogonal set of benefits and drawbacks that can enable experiments where 2P technology is particularly unfit.

Specifically, the sparsity requirement can be satisfied either with sparse expression of the opsin, or, to a certain extent, with sparser photostimulation patterns. The first option is not a conceptual burden for many neuroscience experiments, in fact, the redundancy of the neural code, the hub-like structure of many neural circuits, and the ability of patterns to auto-complete through the network of many neural circuits, and the ability of patterns of activity to auto-complete through the network all allow neuroscientists to usefully and meaningfully address neural circuits even with sparse stimulation. Consider the remarkable results of Carillo-Reid et al., (Cell, 2019 ref. 13) where stimulation a very small population of neurons substantially alters the behaviour of the mouse. Although 2p activation was certainly advantageous, since photo-stimulating such small ensembles can drive major impacts on behaviour, it seems plausible that 3D-MAP, coupled with moderately sparse expression, could be used to execute similar experiments. For instance, even if opsin and calcium indicators were only expressed in ~25% of neurons, it should be possible to impact behaviour with 3D-MAP by identifying and stimulating these specific but highly important neurons. There will be fewer neurons to image/photo-stimulate compared to 2p imaging, but there should still be enough in many cases.

There are other recent 2p holographic studies that also used small numbers of neurons to impact behaviour, including Marshel et al., (ref 46), Dal Maschio et al., (response ref 2), Daie et al., (ref 15), and Gill et al., (ref 20). Although no doubt the higher spatial resolution of 2p optogenetics is useful, it is conceivable, if not likely, that with 3D-MAP and sparse expression (20-30% of cells) one could accomplish similar outcomes on behaviour in many useful applications. Given the much lower cost of 3D-MAP it could bring this type of important experiment into a much broader array of neuroscience labs around the world.

With the second option (sparser illumination), digitally sparsifying holograms by addressing subgroups of neurons, 3D-MAP operates as speeds that are significantly faster than the opsins response. Therefore, this approach should elicit the same response as parallel stimulation with little to no difference in action potential timing. Sparsity requirements are overall a manageable constraint more than a performance-limiting factor in many neuroscience applications. This point in clarified in line 622-634.

In terms of barriers to adoptability in the neuroscience community, 3D-MDP is nearly an order of magnitude less expensive to implement (supplement material – table 2). Therefore, even with the noticeable operational constraints and limitations detailed in our manuscript, our technology remains suitable to explore may applications in a cost-effective manner. We anticipate that many laboratories that already operate 2P systems will consider broadening their capabilities with 3D-MAP, and that many other laboratories that do not have the leverage to purchase 2P technology will consider our approach for a significantly similar price tag.

With regard to spatial resolution, there have been ample one-photon photo-stimulation studies (response ref 3-4) with far lower spatial resolution than 3D-MAP that still provided useful information about the underlying circuits or behaviour. Overall, we think one-photon photo-stimulation and two-photon photo-stimulation are orthogonal solutions that each have the capacity to respond to different experimental needs. Therefore, we think that even if 3D-MAP will only be a suitable replacement to a few applications where 2P technology is considered the gold standard, it will enable a broad range of new applications that even 2P technology is currently unable to address.

1. To clearly show the ability of the system to perform simultaneous calcium imaging and photostimulation the authors should show raw calcium transients of photostimulated and non-photostimulated neurons at different trials, as well as a plot showing calcium transients of non-stimulated neurons that express the opsin as a function of the distance between photostimulated neurons.

We apologize for the lack of clarity. We added a new plot to show calcium transients of Figure 6H (new Figure 7H) as the new Figure 7J.

We also added an example showing calcium transients of non-photostimulated neurons as a function of the distance to the photostimulation location as new Figure 7-sigure supplement 1I, in addition the Figure 7—figure supplement 1C. This result shows the calcium activity of a neuron expressing opsin located as x=0 while the light focus of 3D-MAP sweeps across it in a ±120µm range. We repeated this experiment in 10 L2/3 cells in vivo and the averaged result in the optical PPSF showing in new figure 7E.

2. The manuscript mentioned the cost of budget and high performance realizations of the 1P system but a clear statement of the disadvantages of a budget system is lacking. It would be helpful to include a paragraph indicating what is missing in a budget 1P system compared to a high performance 1P system.

We thank the reviewer for this comment and we added the discussion under Table 2 that compares the cost of a budget system and a high-performance system. For 1P 3D-MAP, the budget system costs 23% of the high-performance system at an expense of performance. First, lasers with lower cost or even LEDs can photo-stimulate opsins, but the maximum power is usually much lower than that of expensive 1P lasers. 3D-MAP distributes the total power from the laser across the whole FOV. To keep the same power density for efficient phot-stimulation, using a lower power laser will result in a smaller FOV for the same sample. Second, sCMOS cameras at lower price may have fewer pixels, higher readout noise, or slower frame rate, which will generate more noisy measurement of calcium signals. Third, a budget DMD has fewer pixels and slower pattern rate (but still above 1 kHz) than a high performance DMD, and budget scanning mirrors have smaller aperture size and scanning range. These factors will limit the total number of accessible voxels, that it, the system will either have low resolution across a large volume or high resolution across a small volume. Finally, the main factor affecting optical table cost will be the thickness of the table, given 3D-MAP is compact in lateral dimension.

3. One of the main issues with all-optical interrogation of neural circuits and behavior is the lack of co-expression of opsins and calcium indicators. This is a crucial step since usually the targeted neurons are the ones that have activity in the behavioral task. However, the manuscript mentions that the system proposed is ideal for sparse expression of opsins, but the authors didn't mention the level of co-expression in their experiments. If the level of co-expression in a sparse labeling is low, then the system will not be ideal for the all-optical interrogation of neural circuits and behavior.

We thank the reviewer for bringing this point to out attention and we added quantitative analysis about the co-expression level to the revised manuscript (line 469-471). We co-expressed CoChR and jRCaMP1a in L2/3 neurons in figure 6. After analyzing a few image stacks, we find the number of neurons expressing CoChR is 114, the number of neurons expressing RCaMP1a is 99, and the number of neurons expressing both is 54. Therefore, the co-expression level is about 50%. We made our best effort to quantify the degree of overlap, although this is likely and underestimate as cells expressing jRCaMP but never firing might not show up as detectably red. Of course, another strategy to ensure 100% co-expression would be to generate a signal bi-cistronic construct co-expressing jRcAMP and CoChR vis a ‘p2A’ signal.

4. The main strength of the paper is the demonstration that the system could be used to map the influence of sparse neurons over a large volume onto a postsynaptic target. So, focusing the rationale, results and conclusions on those experiments could be a better strategy until the 1P system is refined to reach similar performance of a 2P system for broad expression of opsins.

We would like to emphasize the complementarity of the 3D-MAP and 2P photo-stimulation. High spatial resolution and low scattering are the advantages of 2P techniques, but heat and photodamage have already set a fundamental limit that prevents future application in which large populations of neurons must be simultaneously stimulated. Therefore, 2P photo-stimulation is suitable to target deep, but small and close-packed neurons. Conversely, 1P 3D-MAP uses 2-3 orders less leaser power and could potentially simultaneously stimulate 2-3 orders more neurons in a larger area than 2P techniques, but an expense of spatial resolution and penetration depth. Therefore, 3D-MAP is suitable to simultaneously stimulate a large group of neurons in a large volume. In addition, 3-D-MAP is almost one-order less expensive than 2P techniques, even with multiple lasers for a broad range of opsins from blue to red. So we think 3D-MAP could become a complementary technique to 2P techniques for broad applications that are inaccessible with 2P photo-stimulation.

References

M. Dal Maschio, J. C. Donovan, T. O. Helmbrecht, H. Baier, Linking Neurons to Network Function and Behaviour by Two-Photon Holographoic Optogentics and Volumetric Imaging. Neuron. 94, 774-789.e5 (2017).

Adesnik, H., Bruns, W., Tagniguchi, H., Huang, Z. J. and Scanziani, M. A neural circuit for spatial summation in visual cortex. Nature 490, 226-231 (2012)

Lee, S.-H. et al. Activation of specifici interneurons improved V1 feature selectivity and visual perception. Nature 488, 379-383 (2021)